# Cardiovascular Disease-Associated MicroRNAs as Novel Biomarkers of First-Trimester Screening for Gestational Diabetes Mellitus in the Absence of Other Pregnancy-Related Complications

**DOI:** 10.3390/ijms231810635

**Published:** 2022-09-13

**Authors:** Ilona Hromadnikova, Katerina Kotlabova, Ladislav Krofta

**Affiliations:** 1Department of Molecular Biology and Cell Pathology, Third Faculty of Medicine, Charles University, 100 00 Prague, Czech Republic; 2Institute for the Care of the Mother and Child, Third Faculty of Medicine, Charles University, 147 00 Prague, Czech Republic

**Keywords:** cardiovascular microRNAs, early pregnancy, gene expression, gestational diabetes mellitus, prediction, screening, whole peripheral venous blood

## Abstract

We assessed the diagnostic potential of cardiovascular disease-associated microRNAs for the early prediction of gestational diabetes mellitus (GDM) in singleton pregnancies of Caucasian descent in the absence of other pregnancy-related complications. Whole peripheral venous blood samples were collected within 10 to 13 weeks of gestation. This retrospective study involved all pregnancies diagnosed with only GDM (*n* = 121) and 80 normal term pregnancies selected with regard to equality of sample storage time. Gene expression of 29 microRNAs was assessed using real-time RT-PCR. Upregulation of 11 microRNAs (miR-1-3p, miR-20a-5p, miR-20b-5p, miR-23a-3p, miR-100-5p, miR-125b-5p, miR-126-3p, miR-181a-5p, miR-195-5p, miR-499a-5p, and miR-574-3p) was observed in pregnancies destinated to develop GDM. Combined screening of all 11 dysregulated microRNAs showed the highest accuracy for the early identification of pregnancies destinated to develop GDM. This screening identified 47.93% of GDM pregnancies at a 10.0% false positive rate (FPR). The predictive model for GDM based on aberrant microRNA expression profile was further improved via the implementation of clinical characteristics (maternal age and BMI at early stages of gestation and an infertility treatment by assisted reproductive technology). Following this, 69.17% of GDM pregnancies were identified at a 10.0% FPR. The effective prediction model specifically for severe GDM requiring administration of therapy involved using a combination of these three clinical characteristics and three microRNA biomarkers (miR-20a-5p, miR-20b-5p, and miR-195-5p). This model identified 78.95% of cases at a 10.0% FPR. The effective prediction model for GDM managed by diet only required the involvement of these three clinical characteristics and eight microRNA biomarkers (miR-1-3p, miR-20a-5p, miR-20b-5p, miR-100-5p, miR-125b-5p, miR-195-5p, miR-499a-5p, and miR-574-3p). With this, the model identified 50.50% of GDM pregnancies managed by diet only at a 10.0% FPR. When other clinical variables such as history of miscarriage, the presence of trombophilic gene mutations, positive first-trimester screening for preeclampsia and/or fetal growth restriction by the Fetal Medicine Foundation algorithm, and family history of diabetes mellitus in first-degree relatives were included in the GDM prediction model, the predictive power was further increased at a 10.0% FPR (72.50% GDM in total, 89.47% GDM requiring therapy, and 56.44% GDM managed by diet only). Cardiovascular disease-associated microRNAs represent promising early biomarkers to be implemented into routine first-trimester screening programs with a very good predictive potential for GDM.

## 1. Introduction

Gestational diabetes mellitus (GDM), glucose intolerance in pregnancy [1,2,3], increases the risk of the onset of maternal pregnancy-related complications and neonatal morbidity. It also has long-term implications for both mother and child in form of risk of developing type 2 diabetes mellitus and cardiovascular diseases [1,4,5,6].

Several universal screening programs of GDM [1,2,7,8] have been implemented in the routine care of pregnant women. The first screening phase based on the monitoring of a fasting glucose is usually held at first visit during the first trimester of gestation and rules out patients with pre-existing diabetes and detects the occurrence of early GDM. The second screening phase is usually performed at 24–28 weeks of gestation in pregnancies with normal early screening with the oral glucose tolerance test (OGTT) and identifies the occurrence of GDM at the late second and early third pregnancy trimesters. If normal, the OGTT may be repeated again at 32 weeks of gestation [7].

As of now, several promising early predictive models for GDM have been established.

The initial logistic regression model based on the inclusion of maternal characteristics only (maternal age, weight, height, racial origin, family history of diabetes, use of ovulation drugs, birth weight, and previous history of GDM) showed a high accuracy for prediction of GDM at 11–13 weeks of gestation. It reached the following parameters: area under the curve (AUC) 0.823, 95% confidence interval (95% CI) 0.820–0.826, 55.0% sensitivity at a 10.0% false positive rate (FPR) [9]. A slightly older model for the prediction of GDM based on some of the above mentioned factors combined with serum concentrations of adiponectin and sex hormone binding globulin reached similar predictive results (AUC 0.842, 95% CI: 0.817–0.867, 58.6% at a 10.0% FPR) [10].

Similar data were reported by another research group which used a multivariate regression model for the early prediction of GDM. This model was also based on maternal clinical parameters such as age, body mass index (BMI), South/East Asian ethnicity, parity, family history of diabetes, and previous history of GDM (AUC 0.880, 95% CI: 0.850–0.920, 70.2% detection rate at a 10.0% FPR) [11]. Similarly, the same research group later introduced an improved first-trimester risk multivariate prediction model for GDM. This novel model incorporated family history of diabetes, previous history of GDM, South/East Asian ethnicity, parity, BMI, pregnancy-associated plasma protein A (PAPP-A), triglycerides, and lipocalin-2, and achieved a higher discrimination power (AUC 0.910, 95% CI: 0.890–0.960, 76.8% at a 10.0% FPR) [12].

Furthermore, reduced plasma levels of irisin in the first trimester of gestation were implemented into another model based on known risk factors (maternal age, BMI, gestational age at sampling, smoking, ethnicity, pre-existing hypertension or cardiovascular disease, family history of diabetes, physical activity, family history of diabetes, and blood levels of cholesterol, high-density lipoprotein cholesterol, triglycerides, insulin, fasting plasma glucose, and C-reactive protein). This improved the discrimination rate of predicting GDM in a Chinese population (AUC 0.809, 95% CI: 0.763–0.854) [13]. Another independent large-scale study performed in a Chinese population during the first trimester of pregnancy explored a total of 73 variables and also reached a high discriminative power for GDM (AUC 0.800) [14].

An additional non-invasive predictive model consisting of mean arterial blood pressure in the first trimester, age, ethnicity and previous history of GDM demonstrated relatively high predictive ability for a Singaporean population (AUC 0.820, 95% CI: 0.710–0.930), where UK NICE guidelines had poor GDM predictive outcome (AUC 0.600, 95% CI: 0.510–0.700) [15].

Additionally, metabolomics analyses performed on a Japanese population revealed novel promising metabolic biomarkers (serum glutamine, urine ethanolamine, and urine 1,3-diphosphoglycerate). Each biomarker individually demonstrated a high discrimination power for prediction of GDM during the first or early second trimesters of gestation (AUC over 0.800) [16].

First-trimester screening for GDM for an Israeli population reached very high discriminative power in both non-obese women (AUC 0.940, 95% CI: 0.850–0.990, 83.0% at a 10.0% FPR) and obese women (AUC 0.950, 95% CI: 0.880–0.990, 89% at a 10.0% FPR). These screening models were based on the combination of soluble cluster of differentiation 163 (sCD163), tumour necrosis factor alpha (TNFα), placental protein 13 (PP13), and PAPP-A or on the combination of BMI, insulin, sCD163, and TNFα [17].

The latest model was based on maternal clinical characteristics (age and pre-pregnancy BMI); maternal coagulation function (prothrombin time, international standardized ratio, activated partial thromboplastin time, fibrinogen, and thrombin time); and glycolipid metabolism indicators (fasting blood glucose, total cholesterol, triglycerides, low density lipoprotein cholesterol, small and dense low density lipoprotein cholesterol, apolipoprotein B, and apolipoprotein E). This model was applied to a Chinese population in the first trimester of gestation and reached a high clinical value for the prediction of GDM (AUC 0.892, 95% CI: 0.86–0.93) [18].

Previously, the potential usage of coagulation function examination variables such as prothrombin time and activated partial thromboplastin time as novel biomarkers for the prediction of GDM for a Chinese population at 19 weeks of gestation was demonstrated [19].

Similar results were reported for a Chinese population, when a mid-pregnancy risk prediction model for GDM was applied (AUC 0.911, 95% CI: 0.893–0.930). This model was based on maternal status in the combination with ultrasound and serological findings (age, pre-pregnancy BMI, family history of diabetes, polycystic ovary syndrome, previous history of GDM, high systolic pressure, glycosylated haemoglobin levels, triglyceride levels, total cholesterol levels, low density lipoprotein cholesterol levels, C-reactive protein levels, increased subcutaneous fat thickness, and visceral fat thickness) [20].

Similarly, a combined multivariate prediction model performed between 10 and 16 weeks of gestation in an Irish population also achieved a very high level of discrimination for the prediction of GDM (AUC 0.860, 95% CI: 0.774–0.945). This model was based on family history of diabetes, previous perinatal death, overall insulin resistant condition, ultrasound measurements of subcutaneous and visceral abdominal adipose tissue, 8-point skinfold thickness, mid-upper-arm circumference, and weight [21].

Interestingly, the latest study of Eidgahi et al. [22] presented a simplified GDM predictive model with a very good efficiency (AUC 0.83, 95% CI: 0.76–0.90) in an Irani population. This model was based on the mean values of basic indicators (haemoglobin, haematocrit, red blood cell count, and fasting blood glucose) obtained from repeated measures during the first and early second trimesters of gestation. They suggested that this GDM predictive model might be used mainly in poor and low-income countries.

Other models for the early prediction of GDM have not been as effective as the predictive models introduced above [23,24,25,26,27,28,29,30,31,32,33,34,35,36,37,38,39,40].

We focused on the exploration of gene expression profiles of selected cardiovascular disease-associated microRNAs in the whole peripheral venous blood of women during the early stages of gestation. The aim of the study was to assess the predictive potential for GDM in the absence of other pregnancy-related complications.

Previously, by searching the Medline database we identified a large number of microRNAs playing a role in pathogenesis of diabetes mellitus and cardiovascular/cerebrovascular diseases. Finally, we selected a shortlist of 29 microRNAs for the study which have been repeatedly demonstrated by numerous scientific teams to be involved in development and homeostasis of the cardiovascular system, angiogenesis, and adipogenesis. In addition, these microRNAs were reported to be associated with pathological conditions and diseases (vascular endothelial dysfunction and inflammation, hypoxia, hypertension and regulation of hypertension-related genes, obesity, dyslipidaemia, atherosclerosis and atherosclerotic plaque formation, insulin resistance, diabetes mellitus and diabetes-related complications, metabolic syndrome, cardiovascular diseases involving the blood vessels and/or the heart, chronic kidney disease, ischemia/reperfusion injury, cardiac regeneration, and cachexia) (Table 1) [41,42,43,44,45,46,47,48,49,50,51,52,53,54,55,56,57,58,59,60,61,62,63,64,65,66,67,68,69,70,71,72,73,74,75,76,77,78,79,80,81,82,83,84,85,86,87,88,89,90,91,92,93,94,95,96,97,98,99,100,101,102,103,104,105,106,107,108,109,110,111,112,113,114,115,116,117,118,119,120,121,122,123,124,125,126,127,128,129,130,131,132,133,134,135,136,137,138,139,140,141,142,143,144,145,146,147,148,149,150,151,152,153,154,155,156,157,158,159,160,161,162,163,164,165,166,167,168,169,170,171,172,173,174,175,176,177,178,179,180,181,182,183,184,185,186,187,188,189,190,191,192,193,194,195,196,197,198,199,200,201,202,203,204,205,206,207,208,209,210,211,212,213,214,215,216,217,218,219,220,221,222,223,224,225].

The epigenetic profiling of microRNAs (miR-1-3p, miR-16-5p, miR-17-5p, miR-20a-5p, miR-20b-5p, miR-21-5p, miR-23a-3p, miR-24-3p, miR-26a-5p, miR-29a-3p, miR-92a-3p, miR-100-5p, miR-103a-3p, miR-125b-5p, miR-126-3p, miR-130b-3p, miR-133a-3p, miR-143-3p, miR-145-5p, miR-146a-5p, miR-155-5p, miR-181a-5p, miR-195-5p, miR-199a-5p, miR-210-3p, miR-221-3p, miR-342-3p, miR-499a-5p, and miR-574-3p) was the subject of our interest (Table 1).

Up to now, no reports on microRNA gene profiling of the whole peripheral venous blood in early stages of gestation are at disposal in pregnancies with subsequent onset of GDM.

To our knowledge, only several studies have reported promising data on the early diagnosis of GDM during the first trimester of gestation via screening of circulating cardiovascular disease-associated microRNAs in maternal plasma or serum samples [112,130,226,227,228].

## 2. Results

### 2.1. Clinical Characteristics of GDM and Control Pregnancies

The clinical characteristics of GDM and control pregnancies are summarized in Table 2.

From the clinical characteristics of patients, it is obvious that maternal age (mainly advanced maternal age, ≥35 years), BMI (higher BMI values, BMI ≥ 30 kg/m^2^) at early stages of gestation, the necessity to undergo an infertility treatment by assisted reproductive technology, history of miscarriage, the presence of trombophilic gene mutations, positive first-trimester screening for preeclampsia and/or FGR by FMF algorithm, and family history of diabetes mellitus in first-degree relatives represent independent significant risk factors for the subsequent onset of GDM.

### 2.2. Dysregulation of Cardiovascular Disease-Associated MicroRNAs in Early Stages of Gestation in Pregnancies Destinated to Develop GDM

Initially, microRNA gene expression in peripheral blood leukocytes was compared in the early stages of gestation (within 10 to 13 weeks) between pregnancies destinated to develop GDM and term pregnancies with normal course of gestation (Figure 1). Afterwards, early microRNA gene expression was compared between pregnancies destinated to develop GDM and normal term pregnancies with respect to the treatment strategies (GDM pregnancies managed by diet only and GDM pregnancies requiring a combination of diet and administration of appropriate therapy).

Only the data that reached statistical significance after the application of Benjamini–Hochberg correction are discussed below (Appendix A). To interpret the experimental data, new cutoff point *p*-values were set up. Significant results following the Benjamini–Hochberg correction are marked by asterisks for the appropriate significance levels (* for α = 0.05, ** for α = 0.01, and *** for α = 0.001). The data that were statistically non-significant after the application of Benjamini–Hochberg correction (Table 2 and Table 3) are also displayed (Appendix A), but not discussed further.

Upregulation of miR-1-3p (*p* = 0.0028 **), miR-20a-5p (*p* < 0.001 ***), miR-20b-5p (*p* < 0.001 ***), miR-23a-3p (*p* = 0.0065 *), miR-100-5p (*p* < 0.001 ***), miR-125b-5p (*p* = 0.0034 **), miR-126-3p (*p* = 0.0137 *), miR-181a-5p (*p* = 0.0065 *), miR-195-5p (*p* < 0.001 ***), miR-499a-5p (*p* < 0.001 ***), and miR-574-3p (*p* < 0.001 ***) was detected during the first trimester of gestation in pregnancies destinated to develop GDM (Appendix A, Table 3).

MiR-20a-5p (21.49%), miR-20b-5p (18.18%), miR-23a-3p (15.70%), miR-100-5p (20.66%), miR-125b-5p (14.88%), miR-126-3p (14.05%), miR-195-5p (19.83%) miR-499a-5p (14.88%), and miR-574-3p (23.14%) showed moderate sensitivities at a 10.0% FPR to distinguish between normal pregnancies and pregnancies destinated to develop GDM. In contrast, miR-1-3p (12.40%) and miR-181a-5p (10.74%) showed a low sensitivity to differentiate normal pregnancies and pregnancies with subsequent onset of GDM at a 10.0% FPR (Appendix A). This means that the sensitivity in case of miR-1-3p and miR-181a-5p was similar to the false positive rate (10.0%) at which the expression data were assessed.

### 2.3. First-Trimester Combined MicroRNA Screening Is Able to Differentiate between Pregnancies Destinated to Develop GDM and Term Pregnancies with Normal Course of Gestation

Despite the low sensitivities of miR-1-3p (12.40%) and miR-181a-5p (10.74%), the combined screening of all 11 dysregulated microRNA biomarkers (miR-1-3p, miR-20a-5p, miR-20b-5p, miR-23a-3p, miR-100-5p, miR-125b-5p, miR-126-3p, miR-181a-5p, miR-195-5p, miR-499a-5p, and miR-574-3p) showed the highest accuracy for the early identification of pregnancies destinated to develop GDM (AUC 0.742, *p* < 0.001, 63.64% sensitivity, 78.75% specificity, cut off >0.5850). This combined screening identified, in the early stages of gestation, 47.93% of pregnancies destinated to develop GDM at a 10.0% FPR (Figure 2).

### 2.4. The Very Good Accuracy of First-Trimester Combined Screening (MicroRNA Biomarkers and Selected Clinical Characteristics) to Differentiate between Pregnancies Destinated to Develop GDM and Term Pregnancies with Normal Course of Gestation

The effective screening based on the combination of minimal number of basic clinical characteristics (maternal age and BMI at early stages of gestation and an infertility treatment by assisted reproductive technology) and 11 dysregulated microRNA biomarkers (miR-1-3p, miR-20a-5p, miR-20b-5p, miR-23a-3p, miR-100-5p, miR-125b-5p, miR-126-3p, miR-181a-5p, miR-195-5p, miR-499a-5p, and miR-574-3p) showed relatively high accuracy for the early identification of pregnancies destinated to develop GDM (AUC 0.835, *p* < 0.001, 67.50% sensitivity, 92.50% specificity, cut off >0.6929). This combined screening identified, in the early stages of gestation, 69.17% of pregnancies destinated to develop GDM at a 10.0% FPR (Figure 3).

The screening based on the combination of seven clinical characteristics (maternal age and BMI at early stages of gestation, an infertility treatment by assisted reproductive technology, history of miscarriage, the presence of trombophilic gene mutations, positive first-trimester screening for preeclampsia and/or FGR by FMF algorithm, and family history of diabetes mellitus in first-degree relatives) and 11 dysregulated microRNA biomarkers (miR-1-3p, miR-20a-5p, miR-20b-5p, miR-23a-3p, miR-100-5p, miR-125b-5p, miR-126-3p, miR-181a-5p, miR-195-5p, miR-499a-5p, and miR-574-3p) showed the highest possible accuracy for the early identification of pregnancies destinated to develop GDM (AUC 0.869, *p* < 0.001, 72.50% sensitivity, 90.0% specificity, cut off >0.6572). This combined screening identified, in the early stages of gestation, 72.50% of pregnancies destinated to develop GDM at a 10.0% FPR (Figure 4). 

### 2.5. Dysregulation of Cardiovascular Disease-Associated MicroRNAs in Pregnancies Destinated to Develop GDM with Respect to the Treatment Strategies (Diet Only and a Combination of Diet and Administration of Appropriate Therapy)

Concurrently, upregulation of miR-20a-5p (*p* = 0.0015 **, *p* = 0.0098 *), miR-20b-5p (*p* < 0.001 ***, *p* = 0.0054 **), and miR-195-5p (*p* < 0.001 ***, *p* < 0.001 ***) was observed in both groups of pregnancies destinated to develop GDM, irrespective of the treatment strategies (diet only or a combination of diet and therapy).

In addition, upregulation of miR-1-3p (*p* = 0.0045 *), miR-100-5p (*p* = 0.0010 **), miR-125b-5p (*p* = 0.0109 *), miR-499-5p (*p* = 0.0043 *), and miR-574-3p (*p* < 0.001 ***) was observed in only the group of pregnancies destinated to develop GDM, which was managed well by diet only (Appendix A, Table 4).

Sensitivities at a 10.0% FPR were reported for miR-20a-5p (21.78%, 20.0%), miR-20b-5p (15.84%, 30.0%), and miR-195-5p (18.81%, 25.0%) in pregnancies destinated to develop GDM requiring management by diet only or a combination of diet and administration of appropriate therapy.

Sensitivities at a 10.0% FPR were reported for miR-1-3p (13.86%), miR-100-5p (19.80%), miR-125b-5p (14.85%), miR-499a-5p (15.84%), and miR-574-3p (21.78%) in pregnancies destinated to develop GDM requiring diet only (Appendix A).

### 2.6. First-Trimester Combined MicroRNA Screening Is Able to Differentiate between Pregnancies Destinated to Develop GDM Requiring a Combination of Diet and Administration of Appropriate Therapy and Term Pregnancies with Normal Course of Gestation

The combined screening of three microRNA biomarkers (miR-20a-5p, miR-20b-5p and miR-195-5p) in early stages of gestation was able to detect aberrant microRNA expression profile in 30.0% pregnancies destinated to develop GDM requiring a combination of diet and administration of appropriate therapy at a 10.0% FPR (AUC 0.731, *p* < 0.001, 65.0% sensitivity, 73.75% specificity, cut off >0.1987) (Figure 5).

### 2.7. The Very High Accuracy of First-Trimester Combined Screening (MicroRNA Biomarkers and Selected Clinical Characteristics) to Differentiate between Pregnancies Destinated to Develop GDM Requiring a Combination of Diet and Administration of Appropriate Therapy and Term Pregnancies with Normal Course of Gestation

The effective screening based on the combination of minimal number of basic clinical characteristics (maternal age and BMI at early stages of gestation, and an infertility treatment by assisted reproductive technology) and three dysregulated microRNA biomarkers (miR-20a-5p, miR-20b-5p, and miR-195-5p) showed very high accuracy for the early identification of pregnancies destinated to develop GDM requiring a combination of diet and administration of appropriate therapy (AUC 0.949, *p* < 0.001, 89.47% sensitivity, 86.25% specificity, cut off >0.1912). The screening identified 78.95% of cases at a 10.0% FPR in the early stages of gestation (Figure 6).

The screening based on the combination of seven clinical characteristics (maternal age and BMI at early stages of gestation, an infertility treatment by assisted reproductive technology, history of miscarriage, the presence of trombophilic gene mutations, positive first-trimester screening for preeclampsia and/or FGR by FMF algorithm, family history of diabetes mellitus in first-degree relatives) and three dysregulated microRNA biomarkers (miR-20a-5p, miR-20b-5p, and miR-195-5p) showed the highest possible accuracy for the early identification of pregnancies destinated to develop GDM requiring a combination of diet and administration of appropriate therapy (AUC 0.957, *p* < 0.001, 89.47% sensitivity, 90.0% specificity, cutoff >0.2116). This screen identified 89.47% of cases in the early stages of gestation at a 10.0% FPR (Figure 7).

### 2.8. First-Trimester Combined MicroRNA Screening Is Able to Differentiate between Pregnancies Destinated to Develop GDM Managed by Diet Only and Normal Term Pregnancies

The combined screening of eight microRNA biomarkers (miR-1-3p, miR-20a-5p, miR-20b-5p, miR-100-5p, miR-125b-5p, miR-195-5p, miR-499a-5p, and miR-574-3p) was able to detect, in the early stages of gestation, an aberrant microRNA expression profile in 34.65% of pregnancies destinated to develop GDM managed by diet only at a 10.0% FPR (AUC 0.691, *p* < 0.001, 72.28% sensitivity, 60.0% specificity, cut off >0.4980) (Figure 8).

### 2.9. The Very Good Accuracy of First-Trimester Combined Screening (MicroRNA Biomarkers and Selected Clinical Characteristics) to Differentiate between Pregnancies Destinated to Develop GDM Managed by Diet Only and Term Pregnancies with Normal Course of Gestation

The effective screening based on the combination of a minimal number of basic clinical characteristics (maternal age and BMI at early stages of gestation and an infertility treatment by assisted reproductive technology) and eight dysregulated microRNA biomarkers (miR-1-3p, miR-20a-5p, miR-20b-5p, miR-100-5p, miR-125b-5p miR-195-5p, miR-499a-5p, and miR-574-3p) showed relatively good accuracy for the early identification of pregnancies destinated to develop GDM managed by diet only (AUC 0.784, *p* < 0.001, 61.39 sensitivity, 87.50% specificity, cut off >0.6425). This screening identified 50.50% of cases during the early stages of gestation at a 10.0% FPR (Figure 9).

The screening based on the combination of seven clinical characteristics (maternal age and BMI at early stages of gestation, an infertility treatment by assisted reproductive technology, history of miscarriage, the presence of trombophilic gene mutations, positive first-trimester screening for preeclampsia and/or FGR by FMF algorithm, and family history of diabetes mellitus in first-degree relatives) and eight dysregulated microRNA biomarkers (miR-1-3p, miR-20a-5p, miR-20b-5p, miR-100-5p, miR-125b-5p, miR-195-5p, miR-499a-5p, and miR-574-3p) showed the highest possible accuracy for the early identification of pregnancies destinated to develop GDM managed by diet only (AUC 0.835, *p* < 0.001, 77.23% sensitivity, 78.75% specificity, cut off >0.5137. This combined screening identified, in the early stages of gestation, 56.44% of pregnancies destinated to develop GDM managed by diet only at a 10.0% FPR (Figure 10).

### 2.10. Information on MicroRNA-Gene-Biological Pathways Interactions

The KEGG pathway enrichment analysis of 11 microRNAs dysregulated in early stages of gestation in pregnancies destinated to develop GDM revealed a total of 62 pathways, where at least 18 (29.03%) pathways were cancer related. The cancer-related pathways with the highest −ln(*p*-values) were proteoglycans in cancer (hsa05205; 34.738), viral carcinogenesis (hsa05203; 18.144), renal cell carcinoma (hsa05211; 12.364), glioma (hsa05214; 11.400), and pathways in cancer (hsa05200; 11.269).

Other cancer-related pathways showed slightly lower −ln(*p*-values): transcriptional misregulation in cancer (hsa05202; 9.818), chronic myeloid leukaemia (hsa05220; 9.818), non-small cell lung cancer (hsa05223; 9.492), central carbon metabolism in cancer (hsa05230; 9.047), endometrial cancer (hsa05213; 8.698), colorectal cancer (hsa05210; 8.296), thyroid cancer (hsa05216; 7.630), bladder cancer (hsa05219; 7.099), pancreatic cancer (hsa05212; 6.996), acute myeloid leukaemia (hsa05221; 6.648), small cell lung cancer (hsa05222; 5.661), melanoma (hsa05218; 5.424), and choline metabolism in cancer (hsa05231; 4.536) (Figure 11).

The other pathways with the highest −ln(*p*-values) have been shown to play a role in physiological processes and besides the pathogenesis of cancer. These are Hippo signalling pathway (hsa04390; 16.800), adherens junction (hsa04520; 14.198), signalling pathways regulating pluripotency of stem cells (hsa04550; 12.276), p53 signalling pathway (hsa04115; 12.276), and protein processing in endoplasmatic reticulum (hsa04141; 10.769) (Figure 12).

The microRNA/KEGG pathway heatmap and hierarchical clustering demonstrated the level of involvement of particular microRNAs in various biological pathways (Figure 13).

## 3. Discussion

Gene expression of 29 preselected cardiovascular disease-associated microRNAs was compared between pregnancies destinated to develop GDM and normal term pregnancies in the whole peripheral venous blood during the first trimester of gestation. The study was held within the framework of routine screening to assess the risk for a wide array of major fetal chromosomal and non-chromosomal defects as well as other pregnancy-related complications such as PE and/or FGR.

Upregulation of 11 cardiovascular disease-associated microRNAs (miR-1-3p, miR-20a-5p, miR-20b-5p, miR-23a-3p, miR-100-5p, miR-125b-5p, miR-126-3p, miR-181a-5p, miR-195-5p, miR-499a-5p, and miR-574-3p) was detected during the early stages of gestation in the entire group of pregnancies destinated to develop GDM.

To our knowledge, several studies have reported promising data on the early diagnosis of GDM during the first trimester of gestation via screening of circulating microRNAs in maternal plasma/serum or peripheral blood samples. Our study produced similar findings to Yoffe et al. [226], Lamadrid-Romero et al. [130], and Legare et al. [227].

Yoffe et al. validated two upregulated microRNAs (miR-23a and miR-223) as potential plasma biomarkers for early prediction of GDM (after the ninth gestational week and before completion of the 12th week of gestation) in women diagnosed with GDM via a 75 g OGTT performed at 22–24 weeks of gestation [226].

The study of Lamadrid-Romero et al. [130] reported higher miR-125b-5p expression levels in first-trimester serum samples in GDM pregnancies when compared with the control group. On the other hand, the study of Zhang et al. [229] reported downregulation of miR-125b in circulating plasma exosomes in patients with confirmed diagnosis of GDM within 26–40 weeks of pregnancy. Nevertheless, microRNA expression profile may differ between free circulating microRNAs and circulating exosomes; therefore, these findings are not necessarily contradictory results.

Our data may also support the data presented by Tagoma et al. [190], who observed upregulation of miR-100-5p and miR-195-5p in maternal plasma samples collected during the late second and early third pregnancy trimesters in patients who had a positive glucose tolerance test between 23 and 31 weeks of gestation, in which case miR-195-5p showed the highest fold upregulation, similar to our first-trimester study. Our data and the data of Tagoma et al. [190] are also consistent with the data of Wang et al. [230], who also observed increased expression levels of miR-195-5p in serum samples of GDM patients at 25 weeks of gestation.

Concerning miR-20a-5p, our first-trimester data may support the data of Zhu et al. [51] and Cao et al. [52]. Zhu et al. [51] observed upregulation of miR-20a-5p in peripheral blood samples of women at 16–19 weeks of pregnancy, whereas GDM was diagnosed via a 50 g glucose challenge test at 24–28 weeks of pregnancy. Cao et al. [52] observed upregulation of miR-20a-5p in plasma samples derived from patients at the time of diagnosis of GDM determined at 24–28 gestational weeks via performance of 50 g glucose challenge test and 75 g OGTT test.

Nevertheless, our data are inconsistent with the results of other researchers concerning miR-16-5p and miR-17-5p [51,52,228,231]. While in our study, first-trimester whole peripheral blood levels did not differ between pregnancies destinated to develop GDM and control groups, the expression levels of miR-16-5p and miR-17-5p have been reported to be significantly increased in patients with a diagnosis of GDM confirmed at 24–28 gestational weeks [52]. Similarly, Zhu et al. [51], Sorensen et al. [231], and Juchnicka et al. [228] presented similar findings to Cao et al. [52]. Zhu et al. [5] was able to observe upregulation of miR-16-5p and miR-17-5p in peripheral blood samples of women with subsequent onset of GDM at 16–19 weeks of pregnancy. Similarly, Sorensen et al. [231] observed elevated serum levels of miR-16-5p even in the earlier stages of gestation (mean 15th gestational week) in women destinated to develop GDM. Juchnicka et al. [228] showed upregulation of miR-16-5p in first-trimester serum samples of normoglycemic women that developed GDM within the 24–26 gestational weeks.

In addition, Zhao et al. [232] and Sorensen et al. [231] identified miR-29a and miR-29a-3p as other potentially predictive circulating GDM biomarkers. Unfortunately, they did not show any dysregulation when first-trimester expression levels were compared between pregnancies destinated to develop GDM and the control group in our study.

Parallelly, our data concerning miR-155-5p are inconsistent with the study of Wander et al. [112], who observed a positive association between early–mid-pregnancy plasma miR-155-5p levels and occurrence of GDM.

With regard to miR-1-3p, our study produced supportive findings to the study of Kennedy et al. [233], in which they reported increased levels of miR-1-3p in serum extracellular vesicles in patients with confirmed GDM diagnoses within 26–28 gestational weeks that subsequently delivered large-for-gestational-age new-borns (LGA) when compared with appropriately grown-for-gestational-age new-borns (AGA). Nevertheless, our data concerning miR-133a-3p and miR-145-5p are inconsistent with the study of Kennedy et al. [233]. While they observed reduced levels of miR-145-5p and increased levels of miR-133a-3p in GDM pregnancies delivering LGA new-borns, we did not detect any changes in the gene expression of miR-133a-3p and miR-145-5p during the early stages of gestation in pregnancies destinated to develop GDM.

Similarly, our data concerning miR-143-3p and miR-221-3p did not confirm the data of Legare et al. [227], that implemented these first-trimester dysregulated plasmatic microRNAs into the Lasso regression model for prediction of insulin sensitivity estimated by the Matsuda index at the end of the second trimester of pregnancy. However, our data concerning miR-100-5p concurred with Legare et al. [227], who also observed increased levels of miR-100-5p in plasma samples in the early stages of gestation in pregnancies that subsequently developed GDM.

In addition, other studies have introduced a whole range of other circulating microRNAs which were not subject of interest in our study as biomarkers with predictive or diagnostic potential for GDM. These are the following: let-7b-3p [227], miR-10b-5p [227], miR-16-1-3p [227] miR-19a and miR-19b [234], miR-21-3p [53,112], miR-33a-5p [235], miR-130a-3p [227], miR-132 [232], miR-134-5p [231], miR-141-3p [227], miR-142-3p [228], miR-144 [229], miR-144-3p [228], miR-200a-3p [227], miR-205-5p [227], miR-215-5p [227], miR-218-5p [227], miR-222 [232], miR-330-3p [236], miR-338-3p [227], miR-340 [237], miR-375 [227], miR-429 [227], miR-483-5p [227], miR-499a-3p [233], miR-503 [238], miR-512-3p [227], miR-515-5p [227], miR-516a-5p [227], miR-516b-5p [227], miR-517a-3p [227], miR-517b-3p [227], miR-518e-3p [227], miR-518e-5p [227], miR-519a-5p [227], miR-519b-5p [227], miR-519c-5p [227], miR-519d-5p [227], miR-520a-3p [227], miR-520d-3p [227], miR-522-5p [227], miR-523-5p [227], miR-524-3p [227], miR-582-5p [227], miR-873-5p [227], miR-877-5p [227], miR-1283 [227], miR-1323 [239], miR-2116-3p [227], miR-3183 [227], and miR-4772-5p [227].

The current study revealed that aberrant gene expression of miR-1-3p, miR-20a-5p, miR-20b-5p, miR-23a-3p, miR-100-5p, miR-125b-5p, miR-126-3p, miR-181a-5p, miR-195-5p, miR-499a-5p, and miR-574-3p expression is present during the early stages of gestation in pregnancies destinated to develop GDM.

During the first trimester of gestation, we have also recently observed an aberrant expression profile of these cardiovascular disease-associated microRNAs in pregnancies with chronic hypertension (miR-1-3p, miR-20a-5p, and miR-126-3p) and in normotensive pregnancies with subsequent onset of PE (miR-20a-5p, miR-126-3p, miR-181a-5p, and miR-574-3p), FGR (miR-20a-5p, miR-100-5p, miR-181a-5p, miR-195-5p, and miR-574-3p), SGA (miR-1-3p, miR-20a-5p, miR-20b-5p, miR-126-3p, miR-181a-5p, and miR-499a-5p), and/or preterm delivery (miR-20b-5p) [240,241,242].

Parallelly, not long ago we observed the upregulation of 11 microRNAs (miR-1-3p, miR-20a-5p, miR-20b-5p, miR-23a-3p, miR-100-5p, miR-125b-5p, miR-126-3p, miR-181a-5p, miR-195-5p, miR-499a-5p, and miR-574-3p) in the whole peripheral blood samples of mothers with a history of GDM [243]. At the same time, the upregulation of multiple other cardiovascular disease-associated microRNAs (miR-16-5p, miR-17-5p, miR-21-5p, miR-24-3p, miR-26a-5p, miR-29a-3p, miR-103a-3p, miR-130b-3p, miR-133a-3p, miR-143-3p, miR-145-5p, miR-146a-5p, miR-199a-5p, miR-221-3p, and miR-342-3p) was identified postpartum in mothers with a history of GDM [243], which had not yet been present in the early stages of gestation, and probably appeared later with the onset of GDM.

Existing data suggest that dysregulated microRNAs in early pregnancies destinated to develop GDM play a role, not only in the pathogenesis of cardiovascular and cerebrovascular diseases, but also in the pathogenesis of cancer. Since women with a history of GDM were reported to have a higher risk of developing both cardiovascular diseases [244,245,246,247,248] and cancer [249,250,251,252,253,254,255,256], cardiovascular risk assessment [243] together with cancer screening [249] should be implemented into the routine preventive programmes of women with a previous occurrence of GDM.

## 4. Materials and Methods

### 4.1. Patients Cohort

Within the framework of the retrospective case-control study held at the Institute for the Care of Mother and Child, Prague, Czech Republic, within the period 11/2012–5/2018, the whole peripheral venous blood samples were collected at 10–13 gestational weeks from a total of 4187 singleton pregnancies of Caucasian descent. Finally, 3028 out of 4187 pregnancies had complete medical records from the first trimester of gestation until the time of delivery. Out of these 3028 pregnancies, 121 women were consecutively confirmed to only have GDM, where 101 GDM pregnancies were managed by diet only and 20 GDM pregnancies were managed by the combination of diet and therapy (15 patients required insulin administration and metformin was prescribed for 5 patients). GDM was rarely diagnosed during the first trimester of gestation—only in four patients. Otherwise, the onset of GDM was confirmed in majority of patients (*n* = 117) within 24–28 gestational weeks.

Gestational diabetes mellitus was defined as any degree of glucose intolerance with the first onset during gestation [2,3,257]. The International Association of Diabetes and Pregnancy Study Groups’ (IADPSG) recommendations on the diagnosis and classification of hyperglycaemia in pregnancy were followed, and universal early testing was performed in all pregnancies [2]. The first screening phase, during the first trimester of gestation, detected patients with overt diabetes (fasting plasma glucose level ≥ 7.0 mmol/L) and patients with GDM (fasting plasma glucose level ≥ 5.1 mmol/L–<7.0 mmol/L). The second screening phase, 2 h 75 g OGTT at 24–28 weeks of gestation, was performed for all patients not previously found to have overt diabetes or GDM and identified GDM if fasting plasma glucose level was ≥5.1 mmol/L, 1 h plasma glucose was ≥10.0 mmol/L, or 2 h plasma glucose was ≥8.5 mmol/L [2].

Patients newly diagnosed with diabetes mellitus, patients with the occurrence of chronic hypertension, and those carrying growth-restricted or small-for-gestational-age fetuses, or fetuses with anomalies or chromosomal abnormalities were intentionally excluded from the study. Likewise, patients concurrently demonstrating other pregnancy-related complications such as gestational hypertension, preeclampsia, HELLP syndrome, in utero infections, spontaneous preterm birth, preterm prelabour rupture of membranes, fetal demise in utero, or stillbirth were also excluded from the study.

The control group was selected with regard to the uniformity of gestational age at sampling and storage times of biological samples, and included 80 women with normal courses of gestation that delivered healthy infants after the completion of 37 weeks of gestation with a weight above 2500 g.

No woman had a history of any cardiovascular disease (a positive anamnesis of cardiac remodelling, cardiac hypertrophy, heart failure, or acute myocardial infarction). All pregnant women had normal clinical findings (electrocardiography and echocardiography).

### 4.2. Processing of Samples

Homogenized leukocyte lysates were prepared from 200 µL maternal whole peripheral venous blood samples immediately after collection using a QIAamp RNA Blood Mini Kit (Qiagen, Hilden, Germany), according to the manufacturer’s instructions. Firstly, lysis of erythrocytes was performed using EL buffer. Then, the pelleted leukocytes were stored in a mixture of RLT buffer and β-mercaptoethanol (β-ME) at −80 °C for several months until further processing.

Subsequently, a mirVana microRNA Isolation kit (Ambion, Austin, TX, USA) was used to isolate the RNA fraction highly enriched for small RNAs from whole peripheral blood leukocyte lysates.

Concentration and quality of RNA was assessed using a NanoDrop ND-1000 spectrophotometer (NanoDrop Technologies, Wilmington, DE, USA). The A(260/280) absorbance ratio of isolated RNA samples was 1.8–2.1, demonstrating that the RNA samples were pure and could be used for further analysis. The concentration of the isolated RNA ranged within 2.0–10.0 ng/μL.

Real-time RT-PCR analyses were performed regularly every six months to process the collection of frozen samples derived from GDM and normal term pregnancies. The gene expression levels of 29 cardiovascular disease-associated microRNAs (miR-1-3p, miR-16-5p, miR-17-5p, miR-20a-5p, miR-20b-5p, miR-21-5p, miR-23a-3p, miR-24-3p, miR-26a-5p, miR-29a-3p, miR-92a-3p, miR-100-5p, miR-103a-3p, miR-125b-5p, miR-126-3p, miR-130b-3p, miR-133a-3p, miR-143-3p, miR-145-5p, miR-146-5p, miR-155-5p, miR-181a-5p, miR-195-5p, miR-199a-5p, miR-210-3p, miR-221-3p, miR-342-3p, miR-499a-5p, and miR-574-3p) (Table 5) was determined.

mRNAs of the appropriate microRNAs were reverse transcribed into cDNA using a TaqMan MicroRNA assays containing miRNA-specific stem loop primers and a TaqMan MicroRNA Reverse Transcription Kit (Applied Biosystems, Branchburg, NJ, USA). The total reaction volumes were 10 µL. Furthermore, 3 µL of cDNA was mixed with the components of TaqMan MicroRNA assays (specific primers and the TaqMan MGB probes) and the components of the TaqMan Universal PCR Master Mix (Applied Biosystems, Branchburg, NJ, USA). The total reaction volumes were 15 µL. Reverse transcription and real-time qPCR were performed on a 7500 Real-Time PCR System using the TaqMan PCR conditions described in the TaqMan guidelines. The reverse transcription thermal cycling parameters were the following: 30 min at 16 °C, 30 min at 42 °C, 5 min at 85 °C, and then held at 4 °C. The real-time qPCR thermal cycling parameters were the following: 2 min at 50 °C, 10 min at 95 °C, then 50 cycles at 95 °C for 15 s, and 60 °C for 1 min.

Assessment of microRNA gene expression was performed using the comparative Ct method [258]. The geometric mean of the Ct values of selected endogenous controls (RNU58A and RNU38B) was used as a normalization factor [259] to normalize microRNA gene expression. Selection and validation of endogenous controls for microRNA expression studies in whole peripheral blood samples affected by pregnancy-related complications has already been described in one of our previous studies [260]. In brief, the expression of 20 candidate endogenous controls (HY3, RNU6B, RNU19, RNU24, RNU38B, RNU43, RNU44, RNU48, RNU49, RNU58A, RNU58B, RNU66, RPL21, U6 snRNA, U18, U47, U54, U75, Z30, and cel-miR-39) was investigated using NormFinder (NormFinder v.5, Aarhus University Hospital, Aarhus, Denmark) [261]. RNU58A and RNU38B were identified as the most stable small nucleolar RNAs (ncRNA) and equally expressed in patients with normal and abnormal courses of gestation. Therefore, these ncRNA were selected as the most suitable endogenous controls for the normalization of microRNA qPCR expression studies performed on whole peripheral blood samples affected by pregnancy-related complications.

### 4.3. Statistical Analysis

Initially, power analysis was used to determine the sample size required to detect an effect of a given size with a given degree of confidence (G * Power Version 3.1.9.6, Franz Faul, University of Kiel, Kiel, Germany). A total of 51 cases and 51 controls were required to achieve a power of 0.805 and a total of 70 cases and 70 controls were required to achieve a power of 0.902.

With respect to non-normal data distribution, unpaired nonparametric tests were used for subsequent statistical analyses. Initially, microRNA gene expression was compared between GDM and normal term pregnancies using the Mann–Whitney test. Subsequently, microRNA gene expression was compared between particular groups with respect to the treatment strategies using the Kruskal–Wallis one-way analysis of variance. Afterwards, a post-hoc test for comparison between groups and the Benjamini–Hochberg correction were applied [262] (Table 6 and Table 7).

Boxplots display the median, the 75th and 25th percentiles (the upper and lower limits of the boxes), the maximum and minimum values that are no more than 1.5 times the span of the interquartile range (the upper and lower whiskers), outliers (circles), and extremes (asterisks). Statistica software (version 9.0; StatSoft, Inc., Tulsa, OK, USA) was used to produce the boxplots.

Receivers operating characteristic (ROC) curve analyses state the areas under the curves (AUC), *p*-values, the best cutoff point-related sensitivities, specificities, positive and negative likelihood ratios (LR+, LR−) together with the 95% CI (confidence interval). Furthermore, estimated specificities at fixed sensitivities and estimated sensitivities at fixed specificities are stated (MedCalc Software bvba, Ostend, Belgium). Sensitivities at a 90.0% specificity corresponding to a 10.0% false positive rate (FPR) were selected for data presentation. To select the optimal microRNA combinations, logistic regression with subsequent ROC curve analyses were applied (MedCalc Software bvba, Ostend, Belgium).

### 4.4. Information on MicroRNA-Gene-Biological Pathways Interactions

The DIANA miRPath v.3 database (DIANA TOOLS-mirPath v.3 (uth.gr)) and genes union mode were used as an a priori analysis method to perform KEGG pathway enrichment analysis to investigate the regulatory mechanisms of the microRNAs dysregulated in the early stages of gestation in the whole peripheral blood of mothers destinated to develop GDM. The results of this enrichment analysis were expressed as –ln of the *p*-value (−ln(*p*-value)). Preferentially, the database of experimentally verified microRNA targets (Tarbase v7.0) was used. In case that Tarbase v7.0 database did not provide a sufficient list of experimentally verified microRNA targets, the target prediction algorithm (microT-CDS v5.0) was used as an alternative.

In addition, the pathways/categories union mode, an a posteriori analysis method, was applied with the aim to identify merged *p*-values for each pathway significantly enriched with the gene targets of microRNAs dysregulated in early pregnancies destinated to develop GDM. Furthermore, the targeted pathway clusters/heatmap mode was applied to obtain the microRNA/KEGG pathway heatmap with hierarchical clustering.

## 5. Conclusions

Overall, we observed aberrant expression profiles of 11 microRNAs in the whole peripheral venous blood during the first trimester of gestation in pregnancies destinated to develop GDM. We confirmed the observations of other researchers that miR-23a-3p, miR-100-5p, and miR-125b-5p may serve as microRNA biomarkers with early predictive potential for GDM. In addition, novel microRNA biomarkers (miR-1-3p, miR-20a-5p, miR-20b-5p, miR-126-3p, miR-181a-5p, miR-195-5p, miR-499a-5p, and miR-574-3p) were identified, with the potential to predict GDM during the early stages of gestation.

Combined screening of all 11 dysregulated microRNA biomarkers (miR-1-3p, miR-20a-5p, miR-20b-5p, miR-23a-3p, miR-100-5p, miR-125b-5p, miR-126-3p, miR-181a-5p, miR-195-5p, miR-499a-5p, and miR-574-3p) showed the highest accuracy for the early identification of pregnancies destinated to develop GDM irrespective of the severity of the disease. This screening identified, in the early stages of gestation, 47.93% of pregnancies destinated to develop GDM at a 10.0% FPR.

The predictive model for GDM based on microRNA aberrant expression profile was further improved via the implementation of a minimal number of basic clinical characteristics (maternal age and BMI at early stages of gestation and an infertility treatment by assisted reproductive technology). Following this, 69.17% of pregnancies destinated to develop GDM were identified during the early stages of gestation at a 10.0% FPR.

The simplified prediction model for severe GDM (requiring management of diet and administration of appropriate therapy) using the combination of three basic clinical characteristics and three dysregulated microRNA biomarkers (miR-20a-5p, miR-20b-5p, and miR-195-5p) was able to identify 78.95% of cases at a 10.0% FPR during the early stages of gestation.

Parallelly, the simplified prediction model for GDM with a milder course (managed well by diet only) was more complex and required the involvement of three basic clinical characteristics and eight dysregulated microRNA biomarkers (miR-1-3p, miR-20a-5p, miR-20b-5p, miR-100-5p, miR-125b-5p, miR-195-5p, miR-499a-5p, and miR-574-3p). Following this, the model was able to identify 50.50% of cases at a 10.0% FPR during the early stages of gestation.

The implementation of additional clinical variables into the final GDM predictive model is feasible; however, it depends on the availability of the clinical data, which differs between various health care providers.

The screening based on the combination of seven clinical characteristics (maternal age and BMI at early stages of gestation, an infertility treatment by assisted reproductive technology, history of miscarriage, the presence of trombophilic gene mutations, positive first-trimester screening for preeclampsia and/or FGR by FMF algorithm, and family history of diabetes mellitus in first-degree relatives) and microRNA biomarkers showed the highest possible accuracy for the early identification of pregnancies destinated to develop GDM either regardless or with regard to the severity of the disease. The screening was able to identify, in the early stages of gestation, 72.50% of GDM cases in total—89.47% of GDM cases requiring management by diet and administration of appropriate therapy and 56.44% GDM cases managed well by diet only—at a 10.0% FPR. Nevertheless, we prefer to leave the first-trimester GDM screening simplified as much as possible.

The implementation of a novel first-trimester GDM predictive model based on the combination of basic maternal clinical characteristics and aberrant microRNA expression profile into routine screening programmes may significantly improve the care of pregnancies at risk of the development of GDM. In pregnancies identified to be destinated to develop GDM, effective dietary counselling may be already provided during the early stages of gestation, and a healthy-eating plan naturally rich in nutrients and low in fat and calories may be developed to control blood glucose, manage weight, and control heart disease risk factors, such as a high blood pressure and high blood fats. This preventive measure may contribute to lowering the incidence of GDM overall and may also contribute to a reduction in the number of severe GDM cases that require the administration of an appropriate therapy. This may also contribute to a decrease in the occurrence of other pregnancy-related complications such as gestational hypertension, preeclampsia, and fetal growth restriction.

Since women with a history of GDM have an increased risk of developing diabetes (predominantly type 2 diabetes) and cardiovascular diseases later in life, the implementation of effective early screening programme for GDM alongside subsequent preventive measures into early prenatal care may contribute to a subsequent decrease in the occurrence of diabetes and cardiovascular diseases in young and middle-aged mothers. This would also have a large impact on the offspring descending from GDM-affected pregnancies. Accumulating data suggest that exposure to hyperglycemia in utero, as occurs in gestational diabetes mellitus, may expose the offspring to short-term and long-term adverse effects.

The cost of the implementation of the novel first-trimester GDM predictive model based on the combination of basic maternal clinical characteristics and aberrant microRNA expression profile into routine screening programmes is minimal when compared to the costs related to prenatal, peripartal, postpartal, neonatal, postnatal, and lifelong healthcare. In this manner, a significant reduction in healthcare cost can be achieved.

Large-scale follow-up studies need to be performed to verify diagnostic potential of cardiovascular disease-associated microRNA biomarkers to predict the subsequent occurrence of GDM.

Any changes to the epigenome, including the dysregulation of cardiovascular microRNAs induced during the early stages of gestation in pregnancies complicated by GDM, may predispose mothers to later development of diabetes mellitus and cardiovascular/cerebrovascular diseases. This hypothesis may also be supported by our previous finding that epigenetic changes (upregulation of serious cardiovascular microRNAs) appeared in a proportion of women with a history of GDM throughout postpartal life [243].

## 6. Patents

National patent application—Industrial Property Office, Czech Republic (Patent n. PV 2022-335).

## Figures and Tables

**Figure 1 ijms-23-10635-f001:**
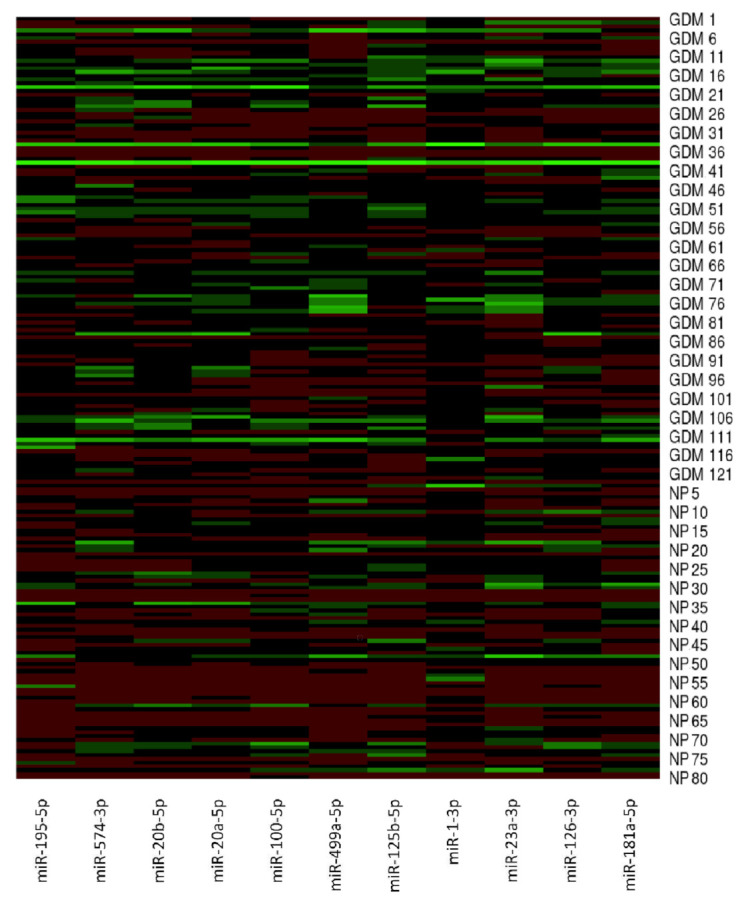
MicroRNA gene expression profile in early stages of gestation in pregnancies destinated to develop GDM and term pregnancies with normal course of gestation. MicroRNA gene expression data (2^−∆∆Ct^) are visualised using the heatmap. In this setting, each row represents a sample (GDM1–GDM121, NP1–NP80) and each column represents a microRNA gene. The colour and intensity of the boxes are used to represent changes of gene expression (2^−∆∆Ct^). Green colour indicates upregulation, and red colour indicates downregulation. GDM; gestational diabetes mellitus, NP; normal pregnancies.

**Figure 2 ijms-23-10635-f002:**
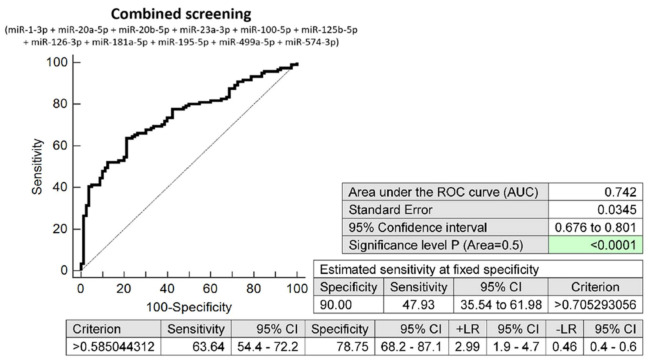
ROC analysis—the combination of microRNA biomarkers (miR-1-3p, miR-20a-5p, miR-20b-5p, miR-23a-3p, miR-100-5p, miR-125b-5p, miR-126-3p, miR-181a-5p, miR-195-5p, miR-499a-5p, and miR-574-3p). A total of 47.93% pregnancies destinated to develop GDM had an aberrant microRNA expression profile in the whole peripheral venous blood during the first trimester of gestation at a 10.0% FPR. This represents 58 out of 121 pregnancies correctly predicted to develop GDM and 8 out of 80 normal pregnancies predicted false positively to develop GDM.

**Figure 3 ijms-23-10635-f003:**
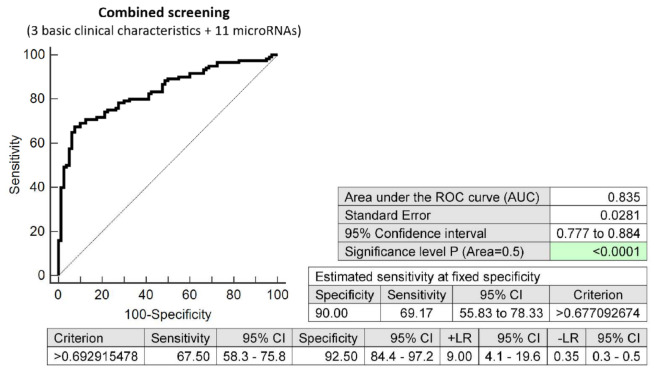
ROC analysis—the combination of 3 basic clinical characteristics (maternal age and BMI values at early stages of gestation and an infertility treatment by assisted reproductive technology) and 11 dysregulated microRNA biomarkers (miR-1-3p, miR-20a-5p, miR-20b-5p, miR-23a-3p, miR-100-5p, miR-125b-5p, miR-126-3p, miR-181a-5p, miR-195-5p, miR-499a-5p, and miR-574-3p). At a 10.0% FPR, 69.17% of pregnancies destinated to develop GDM were identified during the first trimester of gestation. This represents 84 out of 121 pregnancies correctly predicted to develop GDM and 8 out of 80 normal pregnancies predicted false positively to develop GDM.

**Figure 4 ijms-23-10635-f004:**
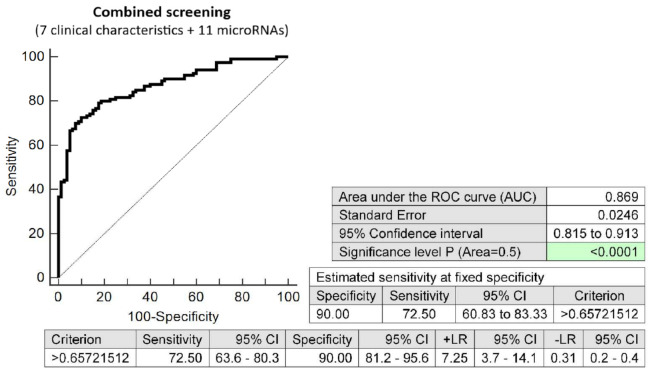
ROC analysis—the combination of 7 clinical characteristics (maternal age and BMI at early stages of gestation, an infertility treatment by assisted reproductive technology, history of miscarriage, the presence of trombophilic gene mutations, positive first-trimester screening for PE and/or FGR by FMF algorithm, and family history of diabetes mellitus in first-degree relatives) and 11 dysregulated microRNA biomarkers (miR-1-3p, miR-20a-5p, miR-20b-5p, miR-23a-3p, miR-100-5p, miR-125b-5p, miR-126-3p, miR-181a-5p, miR-195-5p, miR-499a-5p, and miR-574-3p). At a 10.0% FPR, 72.50% of pregnancies destinated to develop GDM were identified during the first trimester of gestation. This represents 88 out of 121 pregnancies correctly predicted to develop GDM and 8 out of 80 normal pregnancies predicted false positively to develop GDM.

**Figure 5 ijms-23-10635-f005:**
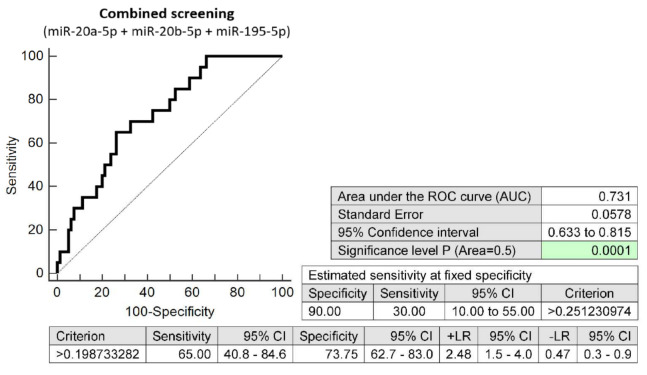
ROC analysis—the combination of microRNA biomarkers (miR-20a-5p, miR-20b-5p and miR-195-5p). A total of 30.0% pregnancies destinated to develop GDM requiring a combination of diet and administration of appropriate therapy had aberrant microRNA expression profile in the whole peripheral venous blood during the first trimester of gestation at a 10.0% FPR. This represents 6 out of 20 pregnancies correctly predicted to develop GDM and 8 out of 80 normal pregnancies predicted false positively to develop GDM.

**Figure 6 ijms-23-10635-f006:**
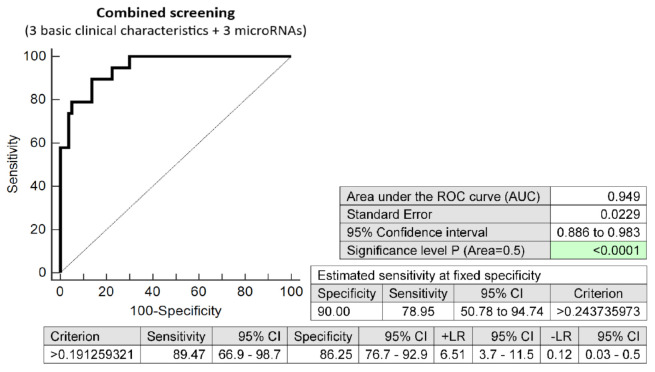
ROC analysis—the combination of 3 basic clinical characteristics (maternal age and BMI values at early stages of gestation and an infertility treatment by assisted reproductive technology) and 3 dysregulated microRNA biomarkers (miR-20a-5p, miR-20b-5p, and miR-195-5p). At a 10.0% FPR, 78.95% pregnancies destinated to develop GDM requiring a combination of diet and administration of appropriate therapy were identified during the first trimester of gestation. This represents 16 out of 20 pregnancies correctly predicted to develop GDM and 8 out of 80 normal pregnancies predicted false positively to develop GDM.

**Figure 7 ijms-23-10635-f007:**
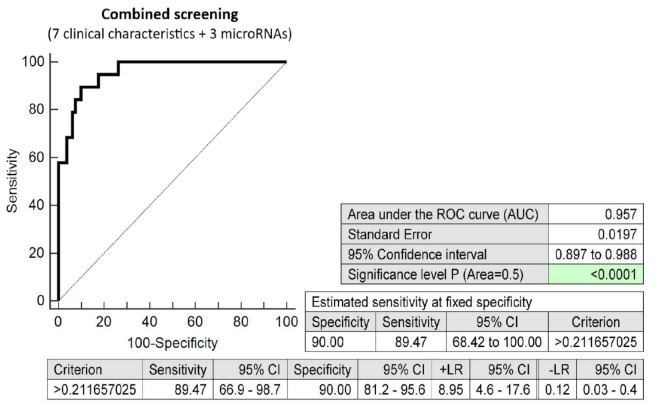
ROC analysis—the combination of 7 clinical characteristics (maternal age and BMI at early stages of gestation, an infertility treatment by assisted reproductive technology, history of miscarriage, the presence of trombophilic gene mutations, positive first-trimester screening for PE and/or FGR by FMF algorithm, and a family history of diabetes mellitus in first-degree relatives) and 3 dysregulated microRNA biomarkers (miR-20a-5p, miR-20b-5p, and miR-195-5p). At a 10.0% FPR, 89.47% pregnancies destinated to develop GDM requiring a combination of diet and administration of appropriate therapy were identified during the first trimester of gestation. This represents 18 out of 20 pregnancies correctly predicted to develop GDM and 8 out of 80 normal pregnancies predicted false positively to develop GDM.

**Figure 8 ijms-23-10635-f008:**
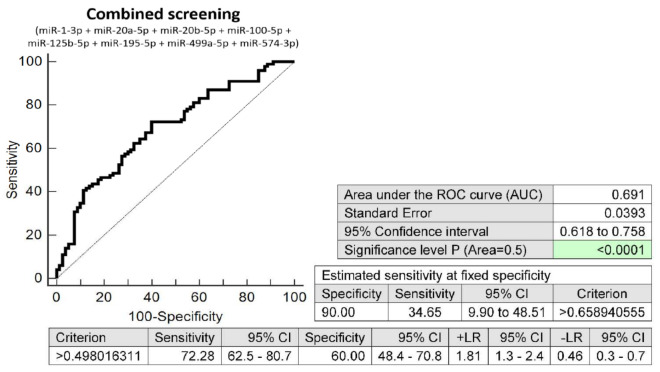
ROC analysis—the combination of microRNA biomarkers (miR-1-3p, miR-20a-5p, miR-20b-5p, miR-100-5p, miR-125b-5p, miR-195-5p, miR-499a-5p, and miR-574-3p). A total of 34.65% pregnancies destinated to develop GDM on diet only had an aberrant microRNA expression profile in the whole peripheral venous blood during the first trimester of gestation at a 10.0% FPR. This represents 35 out of 101 pregnancies correctly predicted to develop GDM and 8 out of 80 normal pregnancies predicted false positively to develop GDM.

**Figure 9 ijms-23-10635-f009:**
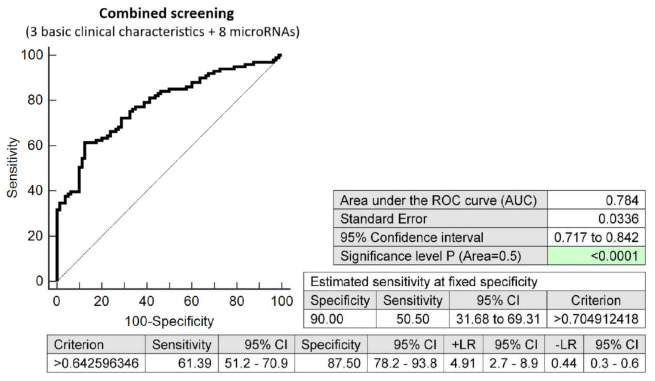
ROC analysis—the combination of 3 basic clinical characteristics (maternal age and BMI values at early stages of gestation and an infertility treatment by assisted reproductive technology) and 8 dysregulated microRNA biomarkers (miR-1-3p, miR-20a-5p, miR-20b-5p, miR-100-5p, miR-125b-5p, miR-195-5p, miR-499a-5p, and miR-574-3p). At a 10.0% FPR, 50.50% pregnancies destinated to develop GDM managed by diet only were identified during the first trimester of gestation. This represents 51 out of 101 pregnancies correctly predicted to develop GDM and 8 out of 80 normal pregnancies predicted false positively to develop GDM.

**Figure 10 ijms-23-10635-f010:**
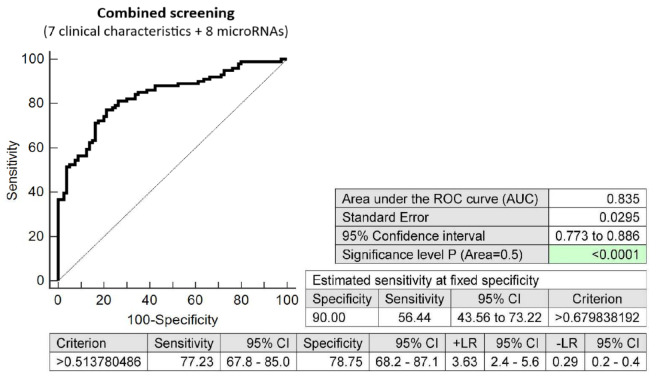
ROC analysis—the combination of 7 clinical characteristics (maternal age and BMI at early stages of gestation, an infertility treatment by assisted reproductive technology, history of miscarriage, the presence of trombophilic gene mutations, positive first-trimester screening for PE and/or FGR by FMF algorithm, and family history of diabetes mellitus in first-degree relatives) and 8 dysregulated microRNA biomarkers (miR-1-3p, miR-20a-5p, miR-20b-5p, miR-100-5p, miR-125b-5p, miR-195-5p, miR-499a-5p, and miR-574-3p). At a 10.0% FPR, 56.44% of pregnancies destinated to develop GDM managed by diet only were identified during the first trimester of gestation. This represents 57 out of 101 pregnancies correctly predicted to develop GDM and 8 out of 80 normal pregnancies predicted false positively to develop GDM.

**Figure 11 ijms-23-10635-f011:**
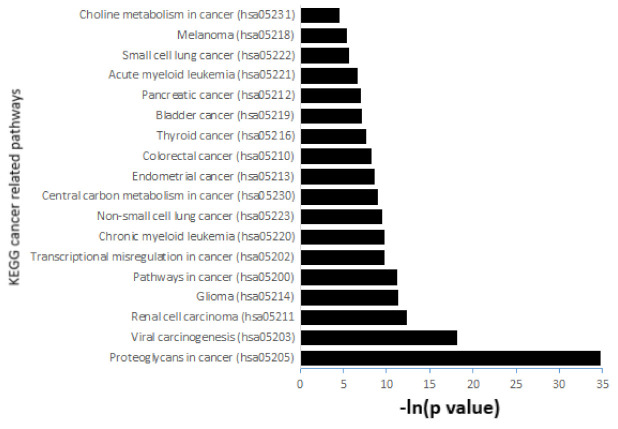
The KEGG pathway enrichment analysis of 11 microRNAs dysregulated in early pregnancies destinated to develop GDM. The analysis revealed a total of 62 pathways, where at least 18 (29.03%) pathways were cancer related. The results were expressed as –ln of the *p*-value (−ln(*p*-value)).

**Figure 12 ijms-23-10635-f012:**
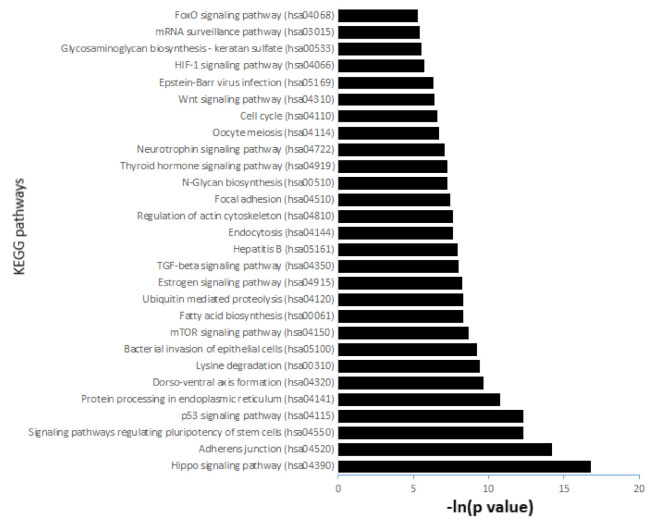
The KEGG pathway enrichment analysis of 11 microRNAs dysregulated in early pregnancies destinated to develop GDM. The analysis revealed a total of 62 various pathways, where a majority of pathways have been shown to play a role in physiological processes and besides to the pathogenesis of cancer. The results were expressed as –ln of the *p*-value (−ln(*p*-value)).

**Figure 13 ijms-23-10635-f013:**
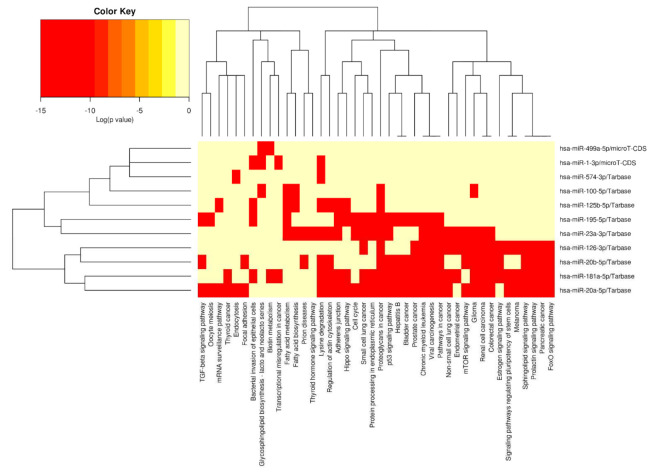
The microRNA/KEGG pathway heatmap and hierarchical clustering in early pregnancies destinated to develop GDM. The heatmap represents the level of involvement of particular microRNAs in various biological pathways. The results were expressed as log of the *p*-value (log(*p*-value)).

**Table 1 ijms-23-10635-t001:** The role of studied microRNAs in the pathogenesis of diabetes mellitus and cardiovascular/cerebrovascular diseases.

miRBase ID	Gene Location on Chromosome	Role in the Pathogenesis of Diabetes Mellitus and Cardiovascular/Cerebrovascular Diseases
hsa-miR-1-3p	20q13.3 [41] 18q11.2	Acute myocardial infarction, heart ischemia, post-myocardial infarction complications, thoracic aortic aneurysm [43], diabetes mellitus [44,45], and vascular endothelial dysfunction [46]
hsa-miR-16-5p	13q14.2	Myocardial infarction [47,48], heart failure [49], acute coronary syndrome, cerebral ischaemic events [50], gestational diabetes mellitus [51,52,53], and diabetes mellitus [54,55,56]
hsa-miR-17-5p	13q31.3 [57,58]	Cardiac development [59], ischemia/reperfusion-induced cardiac injury [60], kidney ischemia-reperfusion injury [61], diffuse myocardial fibrosis in hypertrophic cardiomyopathy [62], acute ischemic stroke [63], coronary artery disease [64], adipogenic differentiation [65], gestational diabetes mellitus [51,52], and diabetes mellitus [56,66]
hsa-miR-20a-5p	13q31.3 [67]	Pulmonary hypertension [68], gestational diabetes mellitus [51,52,69], diabetic retinopathy [70], and diabetes with abdominal aortic aneurysm [71]
hsa-miR-20b-5p	Xq26.2 [67]	Hypertension-induced heart failure [72], insulin resistance [73], T2DM [74,75], and diabetic retinopathy [76]
hsa-miR-21-5p	17q23.2 [77]	Homeostasis of the cardiovascular system [78], cardiac fibrosis and heart failure [79,80], thoracic aortic aneurysm [43], ascending aortic aneurysm [81], regulation of hypertension-related genes [82], myocardial infarction [83], insulin resistance [73], T2DM [84], T2DM with major cardiovascular events [85], T1DM [86,87,88], and diabetic nephropathy [89]
hsa-miR-23a-3p	19p13.12	Heart failure [90], coronary artery disease [91], cerebral ischemia-reperfusion [92], vascular endothelial dysfunction [46], small and large abdominal aortic aneurysm [93], obesity and insulin resistance [94]
hsa-miR-24-3p	19p13.12	Asymptomatic carotid stenosis [95], familial hypercholesterolemia and coronary artery disease [96], angina pectoris [97], ischemic dilated cardiomyopathy [98], small and large abdominal aortic aneurysm [93], myocardial ischemia/reperfusion [99,100], and diabetes mellitus [45,56,60,62]
hsa-miR-26a-5p	3p22.2 [101] 12q14.1	Heart failure, cardiac hypertrophy, myocardial infarction [83,103,104], ischemia/reperfusion injury [105], pulmonary arterial hypertension [106], T1DM [107], and diabetic nephropathy [89]
hsa-miR-29a-3p	7q32.3	Ischemia/reperfusion-induced cardiac injury [108], cardiac cachexia, heart failure [109], atrial fibrillation [110], diffuse myocardial fibrosis in hypertrophic cardiomyopathy [62], coronary artery disease [111], pulmonary arterial hypertension [106], gestational diabetes mellitus [112], and diabetes mellitus [44,55,113,114]
hsa-miR-92a-3p	13q31.3 Xq26.2	Mitral chordae tendineae rupture [115], children with rheumatic carditis [116], myocardial infarction [117], heart failure [118], coronary artery disease [119], and renal injury-associated atherosclerosis [120]
hsa-miR-100-5p	11q24.1	Failing human heart, idiopathic dilated cardiomyopathy, ischemic cardiomyopathy [98], regulation of hypertension-related genes [82], and T1DM [86]
hsa-miR-103a-3p	5q34 [121] 20p13	Hypertension, hypoxia-induced pulmonary hypertension [123], myocardial ischemia/reperfusion injury, acute myocardial infarction [124], ischemic dilated cardiomyopathy [99], obesity, and regulation of insulin sensitivity [125], T1DM [126]
hsa-miR-125b-5p	11q24.1 [126] 21q21.1	Acute ischemic stroke, acute myocardial infarction [128,129], ischemic dilated cardiomyopathy [98], ascending aortic aneurysm [81], gestational diabetes mellitus [130], T1DM [131,132], and T2DM [133]
hsa-miR-126-3p	9q34.3 [134]	Acute myocardial infarction [104], thoracic aortic aneurysm [43], T2DM [85,135], T2DM with major cardiovascular events [85], and gestational diabetes mellitus [136]
hsa-miR-130b-3p	22q11.21	Hypertriglyceridemia [137,138], intracranial aneurysms [139], hyperacute cerebral infarction [140], T2DM [84,141,142], and gestational diabetes mellitus [136]
hsa-miR-133a-3p	18q11.2 [143] 20q13.33	Heart failure, myocardial fibrosis in hypertrophic cardiomyopathy [62,145], arrhythmogenesis in the hypertrophic and failing hearts [146,147], coronary artery calcification [148], thoracic aortic aneurysm [43], ascending aortic aneurysm [81], and diabetes mellitus [41,45]
hsa-miR-143-3p	5q33	Intracranial aneurysms [149], coronary heart disease [150], myocardial infarction [151], myocardial hypertrophy [152], dilated cardiomyopathy [153], pulmonary arterial hypertension [154], acute ischemic stroke [127], and ascending aortic aneurysm [81],
hsa-miR-145-5p	5q33	Hypertension [155,156], dilated cardiomyopathy [157], myocardial infarction [158], stroke [159], acute cerebral ischemic/reperfusion [160], T2DM [56,161], T1DM [84], diabetic retinopathy [162], and gestational diabetes mellitus [163]
hsa-miR-146a-5p	5q33.3 [164,165]	Angiogenesis [166], hypoxia, ischemia/reperfusion-induced cardiac injury [167], myocardial infarction [48], coronary atherosclerosis, coronary heart disease in patients with subclinical hypothyroidism [168], thoracic aortic aneurysm [43], acute ischemic stroke, acute cerebral ischemia [169], T2DM [56,84], T1DM [107], and diabetic nephropathy [89]
hsa-miR-155-5p	21q21.3	Thoracic aortic aneurysm [43], type 1 diabetes [125], gestational diabetes mellitus [53], adolescent obesity [170], diet-induced obesity and obesity resistance [171], atherosclerosis [172], hyperlipidemia-associated endotoxemia [173], coronary plaque rupture [174], children with cyanotic heart disease [175], chronic kidney disease and nocturnal hypertension [176], and atrial fibrillation [177]
hsa-miR-181a-5p	1q32.1 [178] 9q33.3	Regulation of hypertension-related genes, atherosclerosis [178], metabolic syndrome, coronary artery disease [179], non-alcoholic fatty liver disease [180], ischaemic stroke, transient ischaemic attack, acute myocardial infarction [181,182], obesity and insulin resistance [94,178,179], T1DM [84,183], and T2DM [178,182]
hsa-miR-195-5p	17p13.1 [184]	Cardiac hypertrophy, heart failure [185,186], abdominal aortic aneurysms [187], aortic stenosis [188], T2DM [161], and gestational diabetes mellitus [189]
hsa-miR-199a-5p	1q24.3 19p13.2	T1DM, T2DM, gestational diabetes mellitus [190], diabetic retinopathy [191], cerebral ischemic injury [192], heart failure [193], hypertension [194,195], congenital heart disease [196], pulmonary artery hypertension [197], unstable angina [198], hypoxia in myocardium [196], and acute kidney injury [199]
hsa-miR-210-3p	11p15.5	Cardiac hypertrophy [200], acute kidney injury [201], myocardial infarction [202], and atherosclerosis [203]
hsa-miR-221-3p	Xp11.3	Asymptomatic carotid stenosis [95], cardiac amyloidosis [204], heart failure [205], atherosclerosis [206,207], aortic stenosis [208], acute myocardial infarction [209], acute ischemic stroke [210], focal cerebral ischemia [211], pulmonary artery hypertension [212], and obesity [213]
hsa-miR-342-3p	14q32.2	Cardiac amyloidosis [204], obesity [214], T1DM [84,190,215], T2DM [[190],[216],[217], gestational diabetes mellitus [190] and endothelial dysfunction [218]
hsa-miR-499a-5p	20q11.22	Myocardial infarction [48,219], hypoxia [220], cardiac regeneration [221], and vascular endothelial dysfunction [46]
hsa-miR-574-3p	4p14	Myocardial infarction [222], coronary artery disease [138], cardiac amyloidosis [204], stroke [223], and T2DM [142,224]

T1DM: Diabetes mellitus type 1; T2DM: Diabetes mellitus type 2.

**Table 2 ijms-23-10635-t002:** Clinical characteristics of the cases and controls.

	Normal Term Pregnancies (*n* = 80)	GDM Overall (*n* = 121)	GDM Managed by Diet Only (*n* = 101)	GDM Managed by Diet and Therapy (*n* = 20)	*p*-Value ^1^	*p*-Value ^2^	*p*-Value ^3^
*Maternal characteristics*							
Autoimmune diseases (SLE/APS/RA)	0 (0%)	1 (0.83%)	1 (RA, 1.0%)	0 (0%)	0.672 OR: 2.004 95% CI: 0.081–49.814	0.593 OR: 2.403 95% CI: 0.096–59.786	0.497 OR: 3.927 95% CI: 0.076–203.916
Other autoimmune diseases	0 (0%)	1 (0.83%)	1 (vasculitis; 1.0%)	0 (0%)	0.672 OR: 2.004 95% CI: 0.081–49.814	0.593 OR: 2.403 95% CI: 0.096–59.786	0.497 OR: 3.927 95% CI: 0.076–203.916
Any kind of autoimmune disease (SLE/APS/RA/other)	0 (0%)	2 (1.65%)	2 (1.98%)	0 (0%)	0.435 OR: 3.368 95% CI: 0.160–71.088	0.369 OR: 4.045 95% CI: 0.191–85.468	0.497 OR: 3.927 95% CI: 0.076–203.916
Trombophilic gene mutations	0 (0%)	11 (9.09%)	9 (8.91%)	2 (10.0%)	0.052 OR: 16.756 95% CI: 0.973–288.513	0.055 OR: 16.535 95% CI: 0.947–288.589	0.050 OR: 21.757 95% CI: 1.002–472.533
Family history of diabetes							
First-degree relative with DM	10 (12.50%)	30 (24.79%)	26 (25.74%)	4 (20.0%)	0.036 OR: 2.308 95% CI: 1.057–5.037	0.030 OR: 2.427 95% CI: 1.092–5.394	0.392 OR: 1.750 95% CI: 0.486–6.297
Second-degree relative with DM	21 (26.25%)	44 (36.36%)	36 (35.64%)	8 (40.0%)	0.135 OR: 1.605 95% CI: 0.863–2.986	0.178 OR: 1.556 95% CI: 0.818–2.961	0.230 OR: 1.873 95% CI: 0.673–5.215
Parity							
Nulliparous—no previous pregnancy	40 (50.0%)	54 (44.63%)	46 (45.54%)	8 (40.0%)	0.455 OR: 0.806 95% CI: 0.458–1.419	0.551 OR: 0.836 95% CI: 0.465–1.505	0.425 OR: 0.667 95% CI: 0.246–1.805
Parous—no prior GDM	39 (48.75%)	61 (50.41%)	50 (49.50%)	11 (55.0%)	0.817 OR: 1.069 95% CI: 0.608–1.880	0.919 OR: 1.031 95% CI: 0.573–1.853	0.618 OR: 1.285 95% CI: 0.480–3.437
Parous—prior GDM	1 (1.25%)	6 (4.96%)	5 (4.95%)	1 (5.0%)			
History of macrosomia (FBW > 4000 g)	4 (5.0%)	2 (1.65%)	1 (0.99%)	1 (5.0%)	0.194 OR: 0.319 95% CI: 0.057–1.786	0.141 OR: 0.190 95% CI: 0.021–1.735	1.0 OR: 1.000 95% CI: 0.106–9.471
History of miscarriage spontaneous loss of a pregnancy before 22 weeks of gestation	16 (20.0%)	42 (34.71%)	36 (35.64%)	6 (30.0%)	0.026 OR: 2.127 95% CI: 1.095–4.129	0.022 OR: 2.215 95% CI: 1.119–4.384	0.338 OR: 1.714 95% CI: 0.569–5.161
History of perinatal death the death of a baby between 22 weeks of gestation (or weighing 500 g) and 7 days after birth	0 (0%)	4 (3.31%)	3 (2.97%)	1 (5.0%)	0.224 OR: 6.166 95% CI: 0.327–116.113	0.251 OR: 5.721 95% CI: 0.291–112.387	0.128 OR: 12.385 95% CI: 0.486–315.805
ART (IVF/ICSI/other)	2 (2.5%)	20 (16.53%)	15 (14.85%)	5 (25.0%)	0.007 OR: 7.723 95% CI: 1.752–34.038	0.013 OR: 6.802 95% CI: 1.507–30.698	0.004 OR: 13.000 95% CI: 2.304–73.362
Smoking during pregnancy	2 (2.5%)	6 (4.96%)	4 (3.96%)	2 (10.0%)	0.392 OR: 2.035 95% CI: 0.108–10.343	0.589 OR: 1.608 95% CI: 0.287–9.012	0.156 OR: 4.333 95% CI: 0.572–32.859
*Pregnancy details (First trimester of gestation)*							
Maternal age (years)	32 (25–42)	33 (21–42)	33 (21–42)	32 (25–42)	0.635	0.572	0.950
Advanced maternal age (≥35 years old at early stages of gestation)	18 (22.50%)	49 (40.49%)	42 (41.58%)	7 (35.0%)	0.009 OR: 2.618 95% CI: 1.238–4.437	0.007 OR: 2.675 95% CI: 1.271–4.731	0.252 OR: 1.144 95% CI: 0.644–5.343
BMI (kg/m^2^)	21.28 (17.16–29.76)	24.24 (17.37–40.76)	23.89 (17.37–40.76)	26.55 (19.33–39.79)	<0.001	<0.001	<0.001
BMI ≥ 30 kg/m^2^	0 (0%)	25 (20.66%)	17 (16.83%)	8 (40%)	0.009 OR: 42.544 95% CI: 2.550–709.837	0.015 OR: 33.343 95% CI: 1.972–563.719	0.002 OR: 109.480 95% CI: 5.941–2017.344
Gestational age at sampling (weeks)	10.29 (9.57–13.71)	10.29 (9.43–13.57)	10.29 (9.43–13.57)	10.21 (9.43–12.71)	0.737	0.548	0.521
MAP (mmHg)	88.75 (67.67–103.83)	92.0 (72.83–127.58)	91.96 (72.83–127.58)	92.58 (82.85–101.92)	0.051	0.083	0.022
MAP (MoM)	1.05 (0.84–1.25)	1.05 (0.90–1.44)	1.05 (0.90–1.44)	1.07 (0.97–1.13)	0.656	0.574	0.361
Mean UtA-PI	1.39 (0.56–2.43)	1.35 (0.42–2.30)	1.35 (0.42–2.30)	1.25 (0.74–1.84)	0.591	0.831	0.495
Mean UtA-PI (MoM)	0.90 (0.37–1.55)	0.88 (0.26–1.48)	0.89 (0.26–1.48)	0.85 (0.52–1.26)	0.539	0.710	0.402
PIGF serum levels (pg/mL)	27.1 (8.1–137.0)	26.7 (9.2–71.0)	26.8 (9.2–71.0)	25.5 (14.5–46.0)	0.420	0.377	0.375
PIGF serum levels (MoM)	1.04 (0.38–2.61)	1.09 (0.44–2.0)	1.06 (0.44–2.0)	1.15 (0.62–1.59)	0.934	0.690	0.065
PAPP-A serum levels (IU/L)	1.49 (0.48–15.69)	1.28 (0.22–11.45)	1.35 (0.22–11.45)	1.0 (0.26–6.83)	0.063	0.123	0.158
PAPP-A serum levels (MoM)	1.17 (0.37–3.18)	1.05 (1.19–3.67)	1.04 (0.28–3.02)	1.43 (0.19–3.67)	0.606	0.434	0.362
Free b-hCG serum levels (μg/L)	60.21 (9.9–200.6)	50.25 (9.31–211.3)	53.82 (9.31–211.3)	32.62 (16.55–153.2)	0.043	0.123	0.037
Free b-hCG serum levels (MoM)	1.02 (0.31–3.57)	0.98 (0.18–4.54)	1.0 (0.18–4.54)	0.97 (0.33–2.74)	0.317	0.437	0.446
Screen positive for PE and/or FGR by FMF algorithm	0 (0%)	11 (9.09%)	10 (9.90%)	1 (5.0%)	0.052 OR: 16.756 95% CI: 0.973–288.513	0.045 OR: 18.475 95% CI: 1.066–320.312	0.128 OR: 12.385 95% CI: 0.486–315.805
Aspirin intake during pregnancy	0 (0%)	8 (6.61%)	7 (6.93%)	1 (5.0%)	0.089 OR: 12.057 95% CI: 0.686–211.908	0.083 OR: 12.778 95% CI: 0.717–227.208	0.128 OR: 12.385 95% CI: 0.486–315.806
*Pregnancy details (At delivery)*							
BMI (kg/m^2^)	26.66 (21.71–34.82)	28.41 (20.11–49.31)	28.24 (20.11–49.31)	32.11 (23.23–44.98)	0.004	0.042	<0.001
SBP (mmHg)	122 (100–155)	120 (90–160)	121 (90–160)	120 (100–140)	0.823	0.950	0.330
DBP (mmHg)	76 (60–90)	79 (57–109)	79 (57–109)	79 (60–89)	0.898	0.945	0.816
Gestational age at delivery (weeks)	40.07 (37.57–42.0)	39.14 (36.14–41.29)	39.14 (36.14–41.29)	38.93 (36.57–41.0)	<0.001	<0.001	0.009
Delivery at gestational age < 37 weeks	0 (0%)	6 (4.96%)	4 (3.96%) 1 CS for vasculitis-associated adverse obstetric history 3 CS for abnormal CTG	2 (10.0%) 1 CS for vasculitis-associated adverse obstetric history 1 CS for abnormal CTG	0.135 OR: 9.061 95% CI: 0.503–163.118	0.181 OR: 7.431 95% CI: 0.394–140.092	0.050 OR: 21.757 95% CI: 1.002–472.533
Polyhydramnios	1 (1.25%)	28 (23.14%)	21 (20.79%)	7 (35.0%)	0.002 OR: 23.785 95% CI: 3.164–178.781	0.003 OR: 20.738 95% CI: 2.723–157.908	<0.001 OR: 42.538 95% CI: 4.828–374.768
Fetal birth weight (grams)	3470 (2920–4240)	3370 (2430–4340)	3310 (2430–4340)	3625 (2950–4220)	0.043	0.003	0.046
LGA (FBW > 90th percentile)	2 (2.5%)	11 (9.09%)	7 (6.93%)	4 (20.0%)	0.082 OR: 3.900 95% CI: 0.841–18.089	0.192 OR: 2.904 95% CI: 0.586–14.384	0.012 OR: 9.750 95% CI: 1.643–57.851
Macrosomia (FBW > 4000g)	5 (6.25%)	10 (8.26%)	8 (7.92%)	2 (10.0%)	0.596 OR: 1.351 95% CI: 0.444–4.112	0.666 OR: 1.290 95% CI: 0.405–4.108	0.560 OR: 1.667 95% CI: 0.299–9.295
Fetal sex							
Boy	40 (50.0%)	60 (49.59%)	49 (48.51%)	11 (55.0%)	0.954 OR: 0.984 95% CI: 0.559–1.730	0.843 OR: 0.942 95% CI: 0.524–1.695	0.689 OR: 1.222 95% CI: 0.457–3.269
Girl	40 (50.0%)	61 (50.41%)	52 (51.49%)	9 (45.0%)			
Induced delivery	8 (10.0%) 4 postterm pregnancy 1 polyhydramnios 1 suspicious CTG 2 programmed labour	39 (32.23%)	32 (31.68%) 29 term or postterm GDM pregnancy 2 suspicious CTG 1 hepatopathy	7 (35.0%) 7 term or postterm GDM pregnancy	<0.001 OR: 4.281 95% CI: 1.878–9.757	<0.001 OR: 4.174 95% CI: 1.798–9.689	0.008 OR: 4.846 95% CI: 1.498–15.674
Mode of delivery							
Vaginal	69 (86.25%)	66 (54.55%)	58 (57.43%)	8 (40.0%)	<0.001 OR: 5.227 95% CI: 2.519–10.848	<0.001 OR: 4.651 95% CI: 2.199–9.832	<0.001 OR: 9.409 95% CI: 3.139–28.205
CS	11 (13.75%)	55 (45.45%)	43 (42.57%)	12 (60.0%)			
Apgar score < 7, 5 min	0 (0%)	0 (0%)	0 (0%)	0 (0%)	0.837 OR: 0.663 95% CI: 0.013–33.732	0.908 OR: 0.793 95% CI: 0.015–40.411	0.497 OR: 3.927 95% CI: 0.076–203.916
Apgar score < 7, 10 min	0 (0%)	0 (0%)	0 (0%)	0 (0%)	0.837 OR: 0.663 95% CI: 0.013–33.732	0.908 OR: 0.793 95% CI: 0.015–40.411	0.497 OR: 3.927 95% CI: 0.076–203.916
Umbilical blood pH	7.3 (7.29–7.38)		7.3 (7.12–7.39)	7.3 (7.29–7.30)		0.981	0.796

Continuous variables, compared using the Mann–Whitney or Kruskal–Wallis test, are presented as median (range). Categorical variables, presented as number (percent), were compared using odds ratio test. *p*-value ^1,2,3^: the comparison among normal pregnancies and GDM pregnancies, the comparison among normal pregnancies and GDM pregnancies managed by diet only or GDM pregnancies managed by diet and therapy, respectively. GDM, gestational diabetes mellitus; BMI, body mass index; SBP; systolic blood pressure; DBP, diastolic blood pressure; SLE, systemic lupus erythematosus; APS, antiphospholipid syndrome; RA, rheumatoid arthritis; DM, diabetes mellitus; FBW, fetal birth weight; ART, assisted reproductive technology; IVF, in vitro fertilization; ICSI, intracytoplasmic sperm injection; MAP, mean arterial pressure; UtA-PI, uterine artery pulsatility index; PIGF, placental growth factor; PAPP-A, pregnancy-associated plasma protein-A; b-hCG, beta-subunit of human chorionic gonadotropin; PE, preeclampsia; FGR, fetal growth restriction; FMF, Fetal Medicine Foundation; LGA, large for gestational age; CS, caesarean section.

**Table 3 ijms-23-10635-t003:** MicroRNA expression profiles in peripheral blood leukocytes in early stages of gestation in pregnancies destinated to develop GDM and normal term pregnancies.

	Mann-Whitney Test Results GDM Overall (*n* = 121) vs. Normal Term Pregnancies (*n* = 80)		
	Median (IQR)	Mean (SD)	*p*-Value
miR-1-3p	0.135 (0.071–0.254) vs. 0.075 (0.033–0.198)	0.259 (0.525) vs. 0.176 (0.303)	** *p* ** **= 0.0028 ****
miR-16-5p	1.216 (0.968–1.725) vs. 1.411 (0.890–1.980)	1.495 (0.981) vs. 1.646 (1.129)	*p* = 0.5781
miR-17-5p	1.527 (1.181–2.311) vs. 1.384 (0.971–1.923)	1.973 (1.473) vs. 1.748 (1.312)	*p* = 0.0538
miR-20a-5p	2.215 (1.493–3.398) vs. 1.576 (0.991–2.413)	3.037 (3.068) vs. 1.909 (1.370)	***p* < 0.001 *****
miR-20b-5p	2.662 (1.812–3.959) vs. 1.976 (1.111–2.675)	3.706 (3.878) vs. 2.377 (2.291)	***p* < 0.001 *****
miR-21-5p	0.344 (0.231–0.460) vs. 0.320 (0.167–0.538)	0.433 (0.420) vs. 0.394 (0.219)	*p* = 0.2418
miR-23a-3p	0.239 (0.168–0.436) vs. 0.185 (0.103–0.376)	0.367 (0.337) vs. 0.296 (0.329)	***p* = 0.0065 ***
miR-24-3p	0.292 (0.228–0.372) vs. 0.326 (0.196–0.468)	0.331 (0.197) vs. 0.384 (0.284)	*p* = 0.5730
miR-26a-5p	0.699 (0.500–0.926) vs. 0.633 (0.410–1.066)	0.837 (0.670) vs. 0.776 (0.521)	*p* = 0.3022
miR-29a-3p	0.405 (0.282–0.575) vs. 0.372 (0.221–0.545)	0.510 (0.396) vs. 0.407 (0.245)	*p* = 0.0840
miR-92a-3p	2.179 (1.604–3.084) vs. 2.327 (1.188–3.743)	2.702 (2.226) vs. 2.807 (2.132)	*p* = 0.9812
miR-100-5p	0.0023 (0.0013–0.0036) vs. 0.0013 (0.0006–0.0027)	0.0030 (0.0039) vs. 0.0018 (0.0016)	***p* < 0.001 *****
miR-103a-3p	1.565 (0.963–2.541) vs. 1.203 (0.815–2.425)	2.121 (2.252) vs. 1.770 (1.466)	*p* = 0.1547
miR-125b-5p	0.0041 (0.0025–0.0057) vs. 0.0030 (0.0016–0.0054)	0.0049 (0.0046) vs. 0.0036 (0.0027)	***p* = 0.0034 ****
miR-126-3p	0.328 (0.231–0.509) vs. 0.272 (0.140–0.432)	0.462 (0.551) vs. 0.336 (0.270)	***p* = 0.0137 ***
miR-130b-3p	0.745 (0.476–1.409) vs. 0.702 (0.407–1.157)	1.075 (0.960) vs. 1.163 (2.425)	*p* = 0.2105
miR-133a-3p	0.109 (0.061–0.220) vs. 0.110 (0.550–0.233)	0.193 (0.265) vs. 0.232 (0.483)	*p* = 0.8750
miR-143-3p	0.048 (0.030–0.880) vs. 0.038 (0.016–0.089)	0.073 (0.086) vs. 0.058 (0.057)	*p* = 0.0260
miR-145-5p	0.176 (0.125–0.236) vs. 0.161 (0.980–0.243)	0.209 (0.153) vs. 0.195 (0.143)	*p* = 0.2025
miR-146a-5p	1.224 (0.821–1.843) vs. 1.225 (0.578–1.765)	1.658 (1.541) vs. 1.388 (1.096)	*p* = 0.1415
miR-155-5p	0.619 (0.434–0.778) vs. 0.607 (0.361–1.614)	0.703 (0.523) vs 1.247 (1.439)	*p* = 0.2987
miR-181a-5p	0.250 (0.175–0.379) vs 0.181 (0.141–0.330)	0.330 (0.318) vs 0.246 (0.184)	***p* = 0.0065 ***
miR-195-5p	0.267 (0.168–0.487) vs 0.106 (0.048–0.271)	0.470 (0.690) vs 0.227 (0.364)	***p* < 0.001 *****
miR-199a-5p	0.080 (0.037–0.159) vs 0.058 (0.023–0.111)	0.136 (0.223) vs 0.096 (0.131)	*p* = 0.0288
miR-210-3p	0.102 (0.074–0.154) vs 0.138 (0.075–0.224)	0.134 (0.105) vs 0.186 (0.180)	*p* = 0.0952
miR-221-3p	0.644 (0.448–0.969) vs 0.548 (0.293–0.906)	0.815 (0.736) vs 0.693 (0.561)	*p* = 0.0947
miR-342-3p	3.069 (2.122–4.110) vs 2.542 (1.551–4.206)	3.605 (2.724) vs 3.307 (2.383)	*p* = 0.1947
miR-499a-5p	0.460 (0.231–0.780) vs 0.269 (0.089–0.587)	0.758 (1.070) vs 0.477 (0.566)	***p* < 0.001 *****
miR-574-3p	0.275 (0.180–0.395) vs 0.181 (0.117–0.292)	0.354 (0.332) vs 0.222 (0.156)	***p* < 0.001 *****

MicroRNA gene expression is compared between groups using the Mann–Whitney test. Statistically significant results are marked in bold. Median (interquartile range, IQR) and mean (standard deviation, SD) fold values of relative gene expression of samples (2^−∆∆Ct^) are presented. Statistical significant data after Benjamini–Hochberg correction are marked by * for α = 0.05, ** for α = 0.01, and *** for α = 0.001.

**Table 4 ijms-23-10635-t004:** MicroRNA expression profiles in peripheral blood leukocytes in early stages of gestation in pregnancies destinated to develop GDM with respect to the treatment strategies and normal term pregnancies.

	Kruskal–Wallis Test Results GDM Managed by Diet Only (*n* = 101) vs. Normal Term Pregnancies (*n* = 80) GDM Managed by Diet and Therapy (*n* = 20) vs. Normal Term Pregnancies (*n* = 80)		
	Median (IQR)	Mean (SD)	*p*-Value
miR-1-3p	0.141 (0.075–0.274) vs. 0.075 (0.033–0.198) 0.099 (0.071–0.175) vs. 0.075 (0.033–0.198)	0.278 (0.568) vs. 0.176 (0.303) 0.162 (0.190) vs. 0.176 (0.303)	***p* = 0.0045 *** *p* = 1.000
miR-16-5p	1.216 (0.981–1. 785) vs. 1.411 (0.890–1.980) 1.268 (0.923–2.007) vs. 1.411 (0.890–1.980)	1.469 (0.976) vs. 1.646 (1.129) 1.626 (1.019) vs. 1.646 (1.129)	*p* = 1.000 *p* = 1.000
miR-17-5p	1.480 (1.166–2.267) vs. 1.384 (0.971–1.923) 1.893 (1.346–2.362) vs. 1.384 (0.971–1.923)	1.950 (1.553) vs. 1.748 (1.312) 2.085 (0.996) vs. 1.748 (1.312)	*p* = 0.3822 *p* = 0.1019
miR-20a-5p	2.144 (1.486–3.398) vs. 1.576 (0.991–2.413) 2.598 (1.787–3.384) vs. 1.576 (0.991–2.413)	3.019 (3.220) vs. 1.909 (1.370) 3.130 (2.204) vs. 1.909 (1.370)	***p* = 0.0015 **** ***p* = 0.0098 ***
miR-20b-5p	2.577 (1.784–3.719) vs. 1.976 (1.111–2.675) 3.072 (2.085–5.484) vs. 1.976 (1.111–2.675)	3.678 (4.112) vs. 2.377 (2.291) 3.850 (2.439) vs. 2.377 (2.291)	***p* < 0.001 ***** ***p* = 0.0054 ****
miR-21-5p	0.339 (0.222–0.460) vs. 0.320 (0.167–0.538) 0.352 (0.260–0.464) vs. 0.320 (0.167–0.538)	0.426 (0.436) vs. 0.394 (0.219) 0.472 (0.332) vs. 0.394 (0.219)	*p* = 1.000 *p* = 0.4483
miR-23a-3p	0.229 (0.160–0.444) vs. 0.185 (0.103–0.376) 0.299 (0.219–0.344) vs. 0.185 (0.103–0.376)	0.364 (0.346) vs. 0.296 (0.329) 0.383 (0.293) vs. 0.296 (0.329)	*p* = 0.0627 *p* = 0.0371
miR-24-3p	0.292 (0.222–0.370) vs. 0.326 (0.196–0.468) 0.301 (0.241–0.377) vs. 0.326 (0.196–0.468)	0.330 (0.206) vs. 0.384 (0.284) 0.339 (0.147) vs. 0.384 (0.284)	*p* = 1.000 *p* = 1.000
miR-26a-5p	0.729 (0.497–0.938) vs. 0.633 (0.410–1.066) 0.658 (0.560–0.917) vs. 0.633 (0.410–1.066)	0.841 (0.705) vs. 0.776 (0.521) 0.815 (0.462) vs. 0.776 (0.521)	*p* = 0.9599 *p* = 1.000
miR-29a-3p	0.404 (0.276–0.571) vs. 0.372 (0.221–0.545) 0.435 (0.358–0.666) vs. 0.372 (0.221–0.545)	0.486 (0.377) vs. 0.407 (0.245) 0.630 (0.471) vs. 0.407 (0.245)	*p* = 0.5656 *p* = 0.1198
miR-92a-3p	2.171 (1.604–3.036) vs. 2.327 (1.188–3.743) 2.258 (1.603–3.681) vs. 2.327 (1.188–3.743)	2.647 (2.217) vs. 2.807 (2.132) 2.979 (2.3086) vs. 2.807 (2.132)	*p* = 1.000 *p* = 1.000
miR-100-5p	0.0024 (0.0013–0.0036) vs. 0.0013 (0.0006–0.0027) 0.0014 (0.0012–0.0037) vs. 0.0013 (0.0006–0.0027)	0.0031 (0.0041) vs. 0.0018 (0.0016) 0.0028 (0.0025) vs. 0.0018 (0.0016)	***p* = 0.0010 **** *p* = 0.2898
miR-103a-3p	1.531 (0.949–2.533) vs. 1.203 (0.815–2.425) 1.618 (1.234–2.554) vs. 1.203 (0.815–2.425)	2.085 (2.294) vs. 1.770 (1.466) 2.304 (2.075) vs. 1.770 (1.466)	*p* = 0.7368 *p* = 0.4354
miR-125b-5p	0.0041 (0.0026–0.0057) vs. 0.0030 (0.0016–0.0054) 0.0038 (0.0021–0.0055) vs. 0.0030 (0.0016–0.0054)	0.0050 (0.0048) vs. 0.0036 (0.0027) 0.0045 (0.0029) vs. 0.0036 (0.0027)	***p* = 0.0109 *** *p* = 0.4855
miR-126-3p	0.332 (0.219–0.500) vs. 0.272 (0.140–0.432) 0.324 (0.280–0.546) vs. 0.272 (0.140–0.432)	0.470 (0.595) vs. 0.336 (0.270) 0.418 (0.228) vs. 0.336 (0.270)	*p* = 0.0842 *p* = 0.1516
miR-130b-3p	0.707 (0.453–1.315) vs. 0.702 (0.407–1.157) 1.087 (0.577–1.481) vs. 0.702 (0.407–1.157)	1.051 (0.995) vs. 1.163 (2.425) 1.194 (0.769) vs. 1.163 (2.425)	*p* = 1.000 *p* = 0.1983
miR-133a-3p	0.118 (0.066–0.228) vs. 0.110 (0.550–0.233) 0.071 (0.055–0.105) vs. 0.110 (0.550–0.233)	0.209 (0.283) vs. 0.232 (0.483) 0.113 (0.109) vs. 0.232 (0.483)	*p* = 1.000 *p* = 0.4015
miR-143-3p	0.048 (0.029–0.087) vs. 0.038 (0.016–0.089) 0.049 (0.033–0.090) vs. 0.038 (0.016–0.089)	0.072 (0.088) vs. 0.058 (0.057) 0.078 (0.077) vs. 0.058 (0.057)	*p* = 0.1327 *p* = 0.2766
miR-145-5p	0.176 (0.122–0.235) vs. 0.161 (0.980–0.243) 0.171 (0.131–0.242) vs. 0.161 (0.980–0.243)	0.210 (0.162) vs. 0.195 (0.143) 0.200 (0.100) vs. 0.195 (0.143)	*p* = 0.6997 *p* = 1.000
miR-146a-5p	1.116 (0.800–1.798) vs. 1.225 (0.578–1.765) 1.451 (1.167–2.129) vs. 1.225 (0.578–1.765)	1.634 (1.621) vs. 1.388 (1.096) 1.780 (1.068) vs. 1.388 (1.096)	*p* = 0.8676 *p* = 0.1619
miR-155-5p	0.624 (0.432–0.820) vs. 0.607 (0.361–1.614) 0.566 (0.448–0.695) vs. 0.607 (0.361–1.614)	0.701 (0.516) vs. 1.247 (1.439) 0.710 (0.573) vs. 1.247 (1.439)	*p* = 1.000 *p* = 1.000
miR-181a-5p	0.246 (0.175–0.375) vs. 0.181 (0.141–0.330) 0.260 (0.190–0.393) vs. 0.181 (0.141–0.330)	0.331 (0.336) vs. 0.246 (0.184) 0.326 (0.208) vs. 0.246 (0.184)	*p* = 0.0399 *p* = 0.1367
miR-195-5p	0.269 (0.154–0.487) vs. 0.106 (0.048–0.271) 0.246 (0.210–0.522) vs. 0.106 (0.048–0.271)	0.460 (0.707) vs. 0.227 (0.364) 0.520 (0.609) vs. 0.227 (0.364)	***p* < 0.001 ***** ***p* < 0.001 *****
miR-199a-5p	0.073 (0.033–0.139) vs. 0.058 (0.023–0.111) 0.088 (0.052–0.163) vs. 0.058 (0.023–0.111)	0.134 (0.233) vs. 0.096 (0.131) 0.148 (0.165) vs. 0.096 (0.131)	*p* = 0.1575 *p* = 0.1701
miR-210-3p	0.102 (0.074–0.154) vs. 0.138 (0.075–0.224) 0.099 (0.075–0.155) vs. 0.138 (0.075–0.224)	0.134 (0.109) vs. 0.186 (0.180) 0.131 (0.080) vs. 0.186 (0.180)	*p* = 0.2982 *p* = 1.000
miR-221-3p	0.644 (0.448–0.948) vs. 0.548 (0.293–0.906) 0.616 (0.459–1.032) vs. 0.548 (0.293–0.906)	0.819 (0.776) vs. 0.693 (0.561) 0.796 (0.503) vs. 0.693 (0.561)	*p* = 0.3698 *p* = 0.7241
miR-342-3p	3.093 (2.070–3.955) vs. 2.542 (1.551–4.206) 2.884 (2.159–4.844) vs. 2.542 (1.551–4.206)	3.555 (2.756) vs. 3.307 (2.383) 3.858 (2.610) vs. 3.307 (2.383)	*p* = 0.6912 *p* = 1.000
miR-499a-5p	0.459 (0.218–0.881) vs. 0.269 (0.089–0.587) 0.472 (0.285–0.611) vs. 0.269 (0.089–0.587)	0.771 (1.104) vs. 0.477 (0.566) 0.692 (0.902) vs. 0.477 (0.566)	***p*****= 0.0043 *** *p* = 0.1765
miR-574-3p	0.275 (0.182–0.392) vs. 0.181 (0.117–0.292) 0.279 (0.178–0.485) vs. 0.181 (0.117–0.292)	0.350 (0.339) vs. 0.222 (0.156) 0.375 (0.301) vs. 0.222 (0.156)	***p* < 0.001 ***** *p* = 0.0356

MicroRNA gene expression is compared between individual groups using Kruskal–Wallis test. Statistically significant results are marked in bold. Median (interquartile range, IQR) and mean (standard deviation, SD) values of relative fold gene expression of samples (2^−∆∆Ct^) are presented. Statistical significant data after Benjamini–Hochberg correction are marked by * for α = 0.05, ** for α = 0.01, and *** for α = 0.001.

**Table 5 ijms-23-10635-t005:** Characteristics of microRNAs involved in the study.

Assay Name	ID	NCBI Location Chromosome	Sequence
hsa-miR-1	hsa-miR-1-3p	Chr.20: 62554306–62554376 [+]	5′-UGGAAUGUAAAGAAGUAUGUAU-3′
hsa-miR-16	hsa-miR-16-5p	Chr.13: 50048973–50049061 [−]	5′-UAGCAGCACGUAAAUAUUGGCG- 3′
hsa-miR-17	hsa-miR-17-5p	Chr.13: 91350605–91350688 [+]	5′-CAAAGUGCUUACAGUGCAGGUAG-3′
hsa-miR-20a	hsa-miR-20a-5p	Chr.13: 91351065–91351135 [+]	5′-UAAAGUGCUUAUAGUGCAGGUAG-3′
hsa-miR-20b	hsa-miR-20b-5p	Chr.X: 134169809–134169877 [−]	5′-CAAAGUGCUCAUAGUGCAGGUAG-3′
hsa-miR-21	hsa-miR-21-5p	Chr.17: 59841266–59841337 [+]	5′-UAGCUUAUCAGACUGAUGUUGA-3′
hsa-miR-23a	hsa-miR-23a-3p	Chr.19: 13836587–13836659 [−]	5′-AUCACAUUGCCAGGGAUUUCC-3′
hsa-miR-24	hsa-miR-24-3p	Chr.9: 95086021–95086088 [+]	5′-UGGCUCAGUUCAGCAGGAACAG-3′
hsa-miR-26a	hsa-miR-26a-5p	Chr.3: 37969404–37969480 [+]	5′-UUCAAGUAAUCCAGGAUAGGCU-3′
hsa-miR-29a	hsa-miR-29a-3p	Chr.7: 130876747–130876810 [−]	5′-UAGCACCAUCUGAAAUCGGUUA-3′
hsa-miR-92a	hsa-miR-92a-3p	Chr.13: 91351314–91351391 [+]	5′-UAUUGCACUUGUCCCGGCCUGU-3′
hsa-miR-100	hsa-miR-100-5p	Chr.11: 122152229–122152308 [−]	5′-AACCCGUAGAUCCGAACUUGUG-3′
hsa-miR-103	hsa-miR-103a-3p	Chr.5: 168560896–168560973 [−]	5′-AGCAGCAUUGUACAGGGCUAUGA-3′
hsa-miR-125b	hsa-miR-125b-5p	Chr.11: 122099757–122099844 [−]	5′-UCCCUGAGACCCUAACUUGUGA-3′
hsa-miR-126	hsa-miR-126-3p	Chr.9: 136670602–136670686 [+]	5′-UCGUACCGUGAGUAAUAAUGCG-3′
hsa-miR-130b	hsa-miR-130b-3p	Chr.22: 21653304–21653385 [+]	5′-CAGUGCAAUGAUGAAAGGGCAU-3′
hsa-miR-133a	hsa-miR-133a-3p	Chr.18: 21825698–21825785 [−]	5′-UUUGGUCCCCUUCAACCAGCUG-3′
hsa-miR-143	hsa-miR-143-3p	Chr.5: 149428918–149429023 [+]	5′-UGAGAUGAAGCACUGUAGCUC-3′
hsa-miR-145	hsa-miR-145-5p	Chr.5: 149430646–149430733 [+]	5′-GUCCAGUUUUCCCAGGAAUCCCU-3′
hsa-miR-146a	hsa-miR-146a-5p	Chr.5: 160485352–160485450 [+]	5′-UGAGAACUGAAUUCCAUGGGUU-3′
hsa-miR-155	hsa-miR-155-5p	Chr.21: 25573980–25574044 [+]	5′-UUAAUGCUAAUCGUGAUAGGGGU-3′
hsa-miR-181a	hsa-miR-181a-5p	Chr.1: 198859044–198859153 [−]	5′-AACAUUCAACGCUGUCGGUGAGU-3′
hsa-miR-195	hsa-miR-195-5p	Chr.17: 7017615–7017701 [−]	5′-UAGCAGCACAGAAAUAUUGGC-3′
hsa-miR-199a	hsa-miR-199a-5p	Chr.19: 10817426–10817496 [−]	5′-CCCAGUGUUCAGACUACCUGUUC-3′
hsa-miR-210	hsa-miR-210-3p	Chr.11: 568089–568198 [−]	5′-CUGUGCGUGUGACAGCGGCUGA-3′
hsa-miR-221	hsa-miR-221-3p	Chr.X: 45746157–45746266 [−]	5′-AGCUACAUUGUCUGCUGGGUUUC-3′
hsa-miR-342-3p	hsa-miR-342-3p	Chr.14: 100109655–100109753 [+]	5′-UCUCACACAGAAAUCGCACCCGU-3′
mmu-miR-499	hsa-miR-499a-5p	Chr.20: 34990376–34990497 [+]	5′-UUAAGACUUGCAGUGAUGUUU-3′
hsa-miR-574-3p	hsa-miR-574-3p	Chr.4: 38868032–38868127 [+]	5′-CACGCUCAUGCACACACCCACA-3′
RNU58A	664243	Chr.18: 49491283–49491347 [−]	5′-CTGCAGTGATGACTTTCTTGGGACACCTTTGGA TTTACCGTGAAAATTAATAAATTCTGAGCAGC-3′
RNU38B	568914	Chr.1: 44778390–44778458 [+]	5′-CCAGTTCTGCTACTGACAGTAAGTGAAGATAA AGTGTGTCTGAGGAGA-3′

**Table 6 ijms-23-10635-t006:** Benjamini–Hochberg correction: comparison of microRNA gene expression between GDM and normal term pregnancies.

K	i	Alpha = 0.05	Alpha = 0.01	Alpha = 0.001
**2**		**0.05**	**0.01**	**0.001**
	**1**	0.025	0.005	0.001

**Table 7 ijms-23-10635-t007:** Benjamini–Hochberg correction for multiple comparisons: comparison of microRNA gene expression between GDM and normal term pregnancies with respect to the treatment strategies (GDM pregnancies managed by diet only vs. GDM pregnancies managed by diet and therapy vs. normal term pregnancies).

K	i	Alpha = 0.05	Alpha = 0.01	Alpha = 0.001
**3**		**0.05**	**0.01**	**0.001**
	**1**	0.017	0.003	0.000
	**2**	0.033	0.007	0.001
	**3**	0.050	0.010	0.001

## Data Availability

The data presented in this study are available on request from the corresponding author. The data are not publicly available due to rights reserved by funding supporters.

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
