# Peer review of "Cardiovascular Disease-Associated MicroRNAs as Novel Biomarkers of First-Trimester Screening for Gestational Diabetes Mellitus in the Absence of Other Pregnancy-Related Complications"

_ijms, 2022, doi:10.3390/ijms231810635_

Round 1

Reviewer 1 Report

In this manuscript, the authors performed qRT-PCR analysis of selected set of miRNAs to potentially identify diagnostic biomarkers of gestational diabetes mellitus during pregnancy. Although valuable patient samples are used, the molecular characterization of patients is missing. More specific comments are listed below:

Major points:

[1] The writing of the manuscript is poor as most paragraphs only have one sentence. These one-sentences are very long and complicated to be understood.

[2] The rationale for selecting the target miRNAs is weak.

[3] Tables 2 and 3. The numbers of differentially expressed microRNAs must be clearly shown along with their average expression levels in each sample group.

[4] Line 219: “Despite the low sensitivities of miR-1-3p (12.40%) and miR-181a-5p (10.74%)” What do the authors mean by “low sensitivities”?

[5] Due to the lack of negative and positive control gene expression profiling, it is not clear whether the samples/patients reflect the current understanding of dysregulated gene expressions in the conditions that the authors investigated. The authors must characterize the patient cohorts better by providing RNA and protein expression of (i.e., qRT-PCR and immunoblotting, respectively) for marker gene expressions.

Minor points:

(1) The sample size for each experiment is missing in the figure legends.

(2) The primer sequences must be provided.

Reviewer 2 Report

Dear author's

I have reviewed your article entitled" Cardiovascular Disease Associated MicroRNAs as Novel Biomarkers of First Trimester Screening for Gestational Diabetes Mellitus in the Absence of Other Pregnancy-Related Complications" and i have the following comments.

1. The abstract is too long. This section is a short presentation of the study.

2. It is very important to establish the aim of your research and to introduce this info to the section introduction.

3. it will be more accurately if possible to introduce the section methods.

4. Please explain how was established the diagnosis of GDM. (Hear i suggest you to read and cite the new article "Panaitescu AM, Ciobanu AM, Popa M, Duta I, Gica N, Peltecu G, Veduta A. Screening for Gestational Diabetes during the COVID-19 Pandemic-Current Recommendations and Their Consequences. Medicina (Kaunas). 2021 Apr 15;57(4):381. doi: 10.3390/medicina57040381. PMID: 33920937; PMCID: PMC8071285." The diagnosis is very well explained in each trimester.

5. What about the coasts of your method. It is sustainable for a screening test?

6. Please explain the limitation of the method.

Reviewer 3 Report

This study evaluated the role of microRNA evaluation in early pregnancy for the prediction of GDM development. It is an interesting study with clearly presented methodology and results. I have the following comments:

1.      The abstract and introduction section should be shortened.

2.      Why were there only 80 patients selected in the control group? Was this a predefined number?

3.      The authors should further explain the pros and cons of possible implementation of microRNA assessment into the screening programmes. What would be the cost-benefit relationship of this compared to current screening programmes?

Round 2

Reviewer 1 Report

The authors must address further the following previous comments made by this reviewer:

[1] "The sensitivity in case of miR-1-3p (12.4%)  and miR-181a-5p (10.74%) was similar as a false positive rate (10.0%), at which the expression data were assessed." The explanation for how these sensitivity rate was calculated is missing.

[2] Reviewer 1 answer 5

"Since cardiovascular microRNAs are involved in various pathways playing a role in physiological processes, their expression is ubiquitous. They are detectable in peripheral blood, various human tissues inclusive of placental tissue, which was used as a reference sample throughout our studies. There is no suitable human sample that could be used as a negative control, since the selected cardiovascular microRNAs are ubiquitously expressed in human body. As a negative control, no template controls or non-human RNA samples were used in each analysis."

Aside from miRNAs, there are established markers of cardiovascular dysfunctions, such as NPPA, NPPB, Troponin, etc. These must be evaluated in their cohort of patients as it is not clear whether miRNAs the authors identified reflect the disease status without both RNA and protein expression analyses of the known cardiovascular dysfunction markers.

[3] Reviewer 1 answer 7

"The microRNA sequences and location on chromosomes were added to the Methods Section. The sequences of primers are not announced by the producer (Applied Biosystems, Branchburg, USA)."

No information has been provided for internal control primer set.

Reviewer 2 Report

No others suggestion.

Round 3

Reviewer 1 Report

I have no further comment to make.